# Quantifying β-catenin subcellular dynamics and cyclin D1 mRNA transcription during Wnt signaling in single living cells

**Pinhas Kafri[1,2], Sarah E Hasenson[1,2], Itamar Kanter[1,2], Jonathan Sheinberger[1,2], Noa Kinor[1,2], Sharon Yunger[1,2], Yaron Shav-Tal[1,2]***

[1]The Mina and Everard Goodman Faculty of Life Sciences, Bar-Ilan University, Ramat Gan, Israel; [2]Institute of Nanotechnology, Bar-Ilan University, Ramat Gan, Israel

**Abstract** Signal propagation from the cell membrane to a promoter can induce gene expression. To examine signal transmission through sub-cellular compartments and its effect on transcription levels in individual cells within a population, we used the Wnt/β-catenin signaling pathway as a model system. Wnt signaling orchestrates a response through nuclear accumulation of β-catenin in the cell population. However, quantitative live-cell measurements in individual cells showed variability in nuclear β-catenin accumulation, which could occur in two waves, followed by slow clearance. Nuclear accumulation dynamics were initially rapid, cell cycle independent and differed substantially from LiCl stimulation, presumed to mimic Wnt signaling. β-catenin levels increased simultaneously at adherens junctions and the centrosome, and a membrane-centrosome transport system was revealed. Correlating β-catenin nuclear dynamics to *cyclin D1* transcriptional activation showed that the nuclear accumulation rate of change of the signaling factor, and not actual protein levels, correlated with the transcriptional output of the pathway.

***For correspondence:** Yaron.Shav-Tal@biu.ac.il

**Competing interests:** The authors declare that no competing interests exist.

## Introduction

Imaging of gene expression in individual cells using quantitative microscopy has become a central experimental approach for unraveling the dynamic aspects of mRNA transcription (*Darzacq et al., 2009*; *Coulon et al., 2013*; *Hager et al., 2009*), and for examining various events of gene expression in real time (*Darzacq et al., 2007*; *Huranová et al., 2010*; *Brody et al., 2011*; *Martins et al., 2011*; *Yao et al., 2006*; *Mueller et al., 2010*). Cells govern specific transcriptional responses to various stimuli by use of signaling pathways and transducing factors that relay the signal to the promoters of induced target genes (*Purvis and Lahav, 2013*; *Carmo-Fonseca et al., 2002*). Studies of transcription factor dynamics in single cells in response to signaling have revealed dynamic aspects of transcription factor nuclear translocation and modulation (*Kalo and Shav-Tal, 2013*; *Yissachar et al., 2013*; *Lahav et al., 2004*; *Loewer et al., 2010*; *Nelson et al., 2004*; *Vartiainen et al., 2007*). This study centers on the dynamics of the Wnt/β-catenin signaling pathway and its control of *cyclin D1* gene expression, as a model system for examining the dissemination of a signal in the cell and the transcriptional response it elicits.

The Wnt/β-catenin canonical signaling pathway is activated by the binding of the Wnt ligand to plasma membrane receptors, thereby triggering downstream events that culminate in the accumulation of β-catenin in the cytoplasm and its translocation into the nucleus (*Clevers and Nusse, 2012*; *Krieghoff et al., 2006*; *Jamieson et al., 2011*). The interaction of β-catenin with transcription factors

**eLife digest** Cells in an animal's body must communicate with one another to coordinate many processes that are essential to life. One way that cells do this is by releasing molecules that bind to receptors located on the surface of others cells; this binding then triggers a signaling pathway in the receiving cell that passes information from the surface of the cell to its interior. The last stage of these pathways typically involves specific genes being activated, which changes the cell's overall activity.

Wnt is one protein that animal cells release to control how nearby cells grow and divide. One arm of the Wnt signaling pathway involves a protein called β-catenin. In the absence of a Wnt signal, there is little β-catenin in the cell. When Wnt binds to its receptor, β-catenin accumulates and enters the cell's nucleus to activate its target genes. One of these genes, called *cyclin D1*, controls cell division. However it was not understood how β-catenin builds up in response to a Wnt signal and influences the activity of genes.

Using microscopy, Kafri et al. have now examined how the activities of β-catenin and the *cyclin D1* gene change in living human cells. These analyses were initially performed in a population of cells, and confirmed that β-catenin rapidly accumulates after a Wnt signal and that the *cyclin D1* gene becomes activated.

Individual cells in a population can respond differently to signaling events. To assess whether human cells differ in their responses to Wnt, Kafri et al. examined the dynamics of β-catenin in single cells in real time. In most cells, β-catenin accumulated after Wnt activation. However, the time taken to accumulate β-catenin, and this protein's levels, varied between individual cells. Most cells showed the "average" response, with one major wave of accumulation that peaked about two hours after the Wnt signal. Notably, in some cells, β-catenin accumulated in the cell's nucleus in two waves; in other words, the levels in this compartment of the cell increased, dropped slightly and then increased again.

So how does β-catenin in the nucleus activate target genes? Kafri et al. saw that the absolute number of β-catenin molecules in the nucleus did not affect the activity of *cyclin D1*. Instead, cells appeared to sense how quickly the amount of β-catenin in the nucleus changes over time, and this rate influences the activation of *cyclin D1*.

Importantly, problems with Wnt signaling have been linked to diseases in humans; and different components of the Wnt signaling pathway are mutated in many cancers. An important next challenge will be to uncover how the dynamics of this pathway change during disease. Furthermore, a better understanding of Wnt signaling may in future help efforts to develop new drugs that can target the altered pathway in cancer cells.

of the TCF/LEF family in the nucleus modifies gene expression of crucial genes, thus leading to changes in key cellular pathways, such as proliferation, migration and cell fate (*Cadigan and Waterman, 2012*). Mechanistically, in the absence of Wnt, cytoplasmic β-catenin protein is constantly degraded (*Stamos and Weis, 2013*) via the 'destruction complex' and proteosomal degradation (*Aberle et al., 1997*; *Salomon et al., 1997*; *Orford et al., 1997*), thus preventing β-catenin nuclear targeting. In many pathological cases β-catenin is not degraded but accumulates in the nucleus and activates genes, some of which are associated with cell proliferation, such as *MYC* and *cyclin D1* (*Shtutman et al., 1999*; *Tetsu and McCormick, 1999*). The cyclin D1 protein is a major player in the regulation of the cell cycle (*Johnson and Walker, 1999*; *Sherr, 1994*) and its expression is regulated at several levels, including mRNA transcription (*Hosokawa and Arnold, 1998*) via an elaborate promoter region (*Klein and Assoian, 2008*). Cyclin D1 levels were shown to be induced by the Wnt/β-catenin canonical signaling pathway (*Shtutman et al., 1999*; *Tetsu and McCormick, 1999*; *Chocarro-Calvo et al., 2013*; *Willert et al., 2002*; *Lin et al., 2000*; *Porfiri et al., 1997*; *Yun et al., 2005*; *Torre et al., 2011*).

The Wnt/β-catenin signaling pathway has received much experimental attention due to its centrality in gene expression patterning, and its involvement in many cancer types (*Klaus and Birchmeier, 2008*). While the endpoint of β-catenin protein stabilization by Wnt signaling has been well studied

biochemically, the kinetic aspects of this signaling pathway in living cells, for the β-catenin protein and the target mRNA, remain under-studied. To address this topic we used a cell system for the in vivo visualization and analysis of the mammalian mRNA transcriptional kinetics of single alleles (*Yunger et al., 2010*, *2013*). Whereas, we had previously followed transcription from a single *cyclin D1* (*CCND1*) gene in living human cells, we now set out to examine the real-time behavior of β-catenin during active signaling in a population of living cells, and the effect of signaling on the activity pattern of the target gene.

## Results

### System for studying Wnt/β-catenin signaling and gene activation in single living cells

We previously generated a cell system in which a *CCND1* gene was integrated as a single copy allele into human HEK293 cells using Flp-In recombination (*Yunger et al., 2010*). Transcription kinetics on this gene were visualized and quantified using RNA FISH and live-cell imaging techniques. RNA tagging was achieved using a series of MS2 sequence repeats (*Bertrand et al., 1998*) inserted into the long 3'UTR of *CCND1*. The MS2 repeats form stem-loop structures in the transcribed mRNA. By co-expressing a fluorescent coat protein termed MS2-CP-GFP that binds to the MS2 stem-loops, we obtained fluorescent tagging of the mRNAs produced from this gene, designated *CCND1-MS2* (*Yunger et al., 2010*, *2013*). This *CCND1-MS2* allele is under the regulation of the endogenous *cyclin D1* promoter (*Albanese et al., 1995*) and therefore serves as a candidate gene for activation by Wnt/β-catenin signaling (*Fu et al., 2004*).

Studying individual living cells, we found that the *CCND1-MS2* gene transits between transcriptionally active and non-active states under steady-state conditions (*Yunger et al., 2010*). At steady state, only around 40–50% of the cells were actively transcribing *CCND1-MS2*. In order to verify that the Wnt signaling pathway activates the *CCND1-MS2* gene we added Wnt3a conditioned medium to the cells and imaged the cells over time. Indeed, on the population level, after 75 min over 80% of cells had shown an actively transcribing *CCND1-MS2* gene (*Figure 1a,b*, *Supplementary file 1a*, *Video 1*).

Since an imaging-based approach for studying signaling dynamics requires that relevant molecules be fluorescently tagged, we verified using a luciferase assay, that a YFP-tagged version of β-catenin (*Krieghoff et al., 2006*) activates the *CCND1* promoter, and observed 2.3 fold activation after transient transfection of the protein into the HEK293 *CCND1-MS2* cells (*Figure 1—figure supplement 1a*). We note that HEK293 cells are known to have a low background of β-catenin activity (*Kang et al., 2012*), and are not known to have mutations in proteins associated with Wnt signaling (*Tan et al., 2012*). Immunofluorescence with an antibody to the endogenous β-catenin protein showed normal β-catenin localization at the cell membrane region (a portion of β-catenin is located in adherens junctions and functions in cell adhesion [*Harris and Tepass, 2010*]), as well as low cytoplasmic levels under non-induced conditions, compared to a predominant increase in cytoplasmic and nuclear distribution after the addition of Wnt3a (*Figure 1c*). In summary, this cell system enables the measurement of *CCND1* transcription activation kinetics in single cells following Wnt signaling.

To mimic endogenous β-catenin distribution using YFP-β-catenin, we generated a HEK293 *CCND1-MS2* cell clone that stably expressed YFP-β-catenin. Since high overexpression conditions of YFP-β-catenin typically result in increased subcellular distribution and high accumulation in the nucleus prior to any signal (*Figure 1—figure supplement 1b*), which is in stark contrast to the endogenous β-catenin protein that is observed mainly at the membrane (*Figure 1c*), we screened and identified a clone that stably expressed low levels of YFP-β-catenin. The clone phenotypically resembled endogenous protein localization and distribution, namely, membrane localization in the non-induced state, and enhanced nuclear localization following Wnt stimulation (*Figure 1d*). Characterization of endogenous β-catenin and YFP-β-catenin accumulation levels by Western blotting showed that YFP-β-catenin expression levels were ~80% of the endogenous β-catenin, thus doubling β-catenin levels in the cell, and that the accumulation dynamics of both proteins were identical (*Figure 1e*). The time-scale of β-catenin induction is in agreement with other studies (*Hernández et al., 2012*; *Lustig et al., 2002*). The addition of YFP-β-catenin to the cell clone did

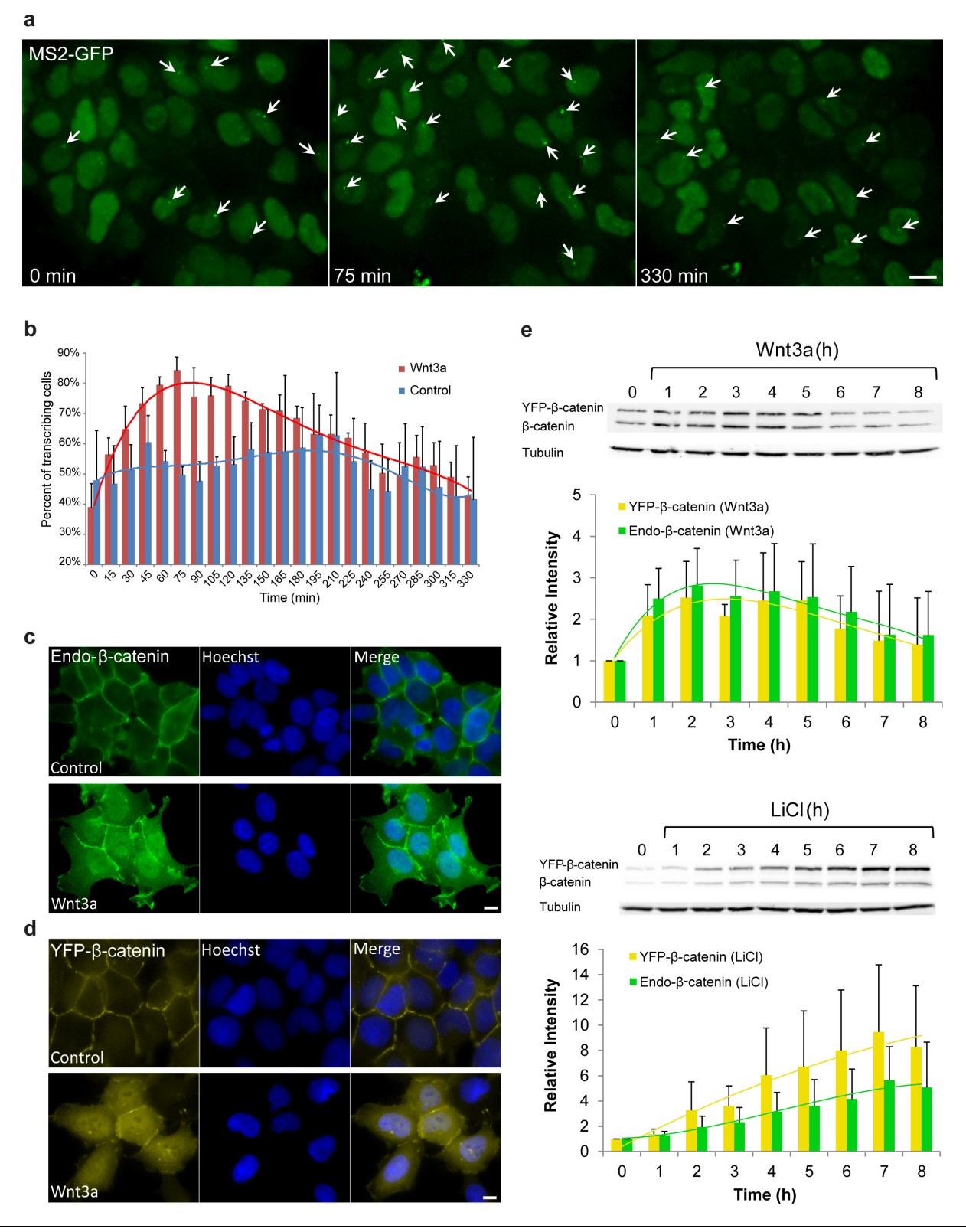

**Figure 1.** Cell system for following β-catenin intra-cellular dynamics and *CCND1* transcription in single living cells. (**a**) CCND1-MS2 HEK293 cells stably expressing MS2-GFP-CP were treated with Wnt3a and followed for 6 hr (every 15 min). Several frames from *Video 1* are presented. The number of cells exhibiting transcriptionally active CCND1-MS2 genes (green dot in the nucleus, white arrow) was counted over time. Scale bar, 10 μm. (**b**) Plots showing the percentage of cells in the population with actively transcribing CCND1-MS2 genes in Wnt3a-treated (red, n = 98) and mock treated (blue, n = 128)

*Figure 1 continued on next page*

*Figure 1 continued*

cells. Mean ± sd from three fields imaged on different days—see **Supplementary file 1a** for statistics. (c) Immunofluorescence showing that endogenous β-catenin (green) is prominent at the cell membrane in untreated HEK293 cells (top) and accumulates in the cytoplasm and nucleus following activation by Wnt3a for 2 hr (bottom). Hoechst DNA stain is in blue. (d) Similar changes in the subcellular distribution following activation are seen in the YFP-β-catenin low-expressing clone. Bar = 10 μm. (e) Western blot time course of endogenous β-catenin and YFP-β-catenin protein accumulation following either Wnt3a (top) or LiCl (bottom) stimulation. Anti-β-catenin antibody was used for the detection of both β-catenin proteins. Tubulin was used as a loading control. Time 0 is the time point of activator addition. Blots are representative of 3 repeated experiments. The average quantification of 3 repeated experiments is presented in the plots below (mean ± sd). There is no statistical difference between the endogenous and exogenous levels of β-catenin in the two plots.

The following figure supplement is available for figure 1:

**Figure supplement 1.** Measuring the effect of YFP-β-catenin expression in HEK293 cells.

not influence the cell cycle or *CCND1* expression at steady state as quantified by single molecule RNA FISH (*Yunger et al., 2010,2013*) (*Figure 1—figure supplement 1c–h*).

## Real-time β-catenin dynamics in a cell population in response to Wnt signaling

To understand the intra-cellular dynamics of β-catenin in a cell population under living cell conditions, cells were imaged for over 12 hr. Rapid nuclear accumulation of β-catenin was observed in most cells that were stimulated with Wnt3a, compared to no change in β-catenin levels in control cells that received mock conditioned medium without Wnt3a (*Figure 2a,b*, *Video 2*). Rising levels of β-catenin in the cytoplasm and the nucleus were detected 15 min after Wnt3a addition, and the accumulation peak was observed 2–3 hr later (*Figure 2c*), during which β-catenin levels increased 3-fold compared to the initial state. Recombinant Wnt3a (200 ng/ml) showed the same dynamics (data not shown). The rate at which β-catenin levels increased in the nucleus was faster than in the cytoplasm, leading to a higher nucleus/cytoplasm (N/C) protein ratio, whereas in the control cells there was no change (*Figure 2d*).

Analyzing the rate of change in β-catenin levels in the nucleus and cytoplasm over time (ΔI/Δt) showed that the accumulation was comprised of two phases; an initial rapid one, in which the peak of the change in accumulation was reached 60 min after induction, and a second accumulation phase in which cellular β-catenin continued to amass but at a declining rate up until 180 min (*Figure 2e*). Subsequently, the rate of change turned negative, meaning that β-catenin levels were declining, probably due to degradation. In control cells, the rate of change in β-catenin remained unaltered.

To examine whether the dynamics of nuclear entry of β-catenin were modified during Wnt activation and how they compared to β-catenin shuttling out of the nucleus, we used fluorescence recovery after photobleaching (FRAP). Nuclei of cells showing nuclear β-catenin, either after 2 hr of Wnt3a activation or transiently over-expressing β-catenin, were photobleached, and nuclear import of β-catenin was monitored over time (*Figure 2—figure supplement 1a* top). The dynamics were relatively slow, however, the import rate under Wnt3a conditions was more rapid than transient overexpression, showing the advantage of measurements performed at low

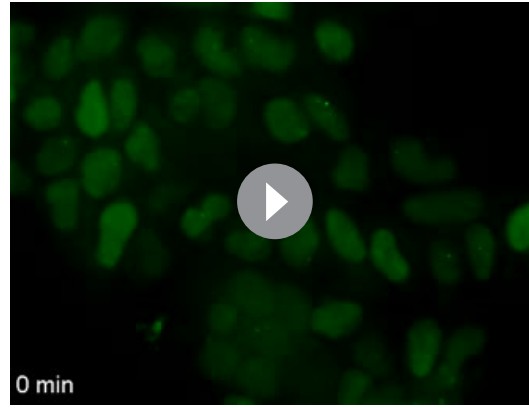

**Video 1.** Transcriptional activation of CCND1 in response to Wnt3a. HEK293 CCND1-MS2 cells stably expressing MS2-GFP (green) were treated with Wnt3a. The transcribed CCND1 mRNA on the active gene is seen as a bright green dot. The fluorescent signal on the active genes was enhanced using ImageJ 'Spot Enhancing Filter 2D' in order to clearly present the active sites in the movie. Cells were imaged every 15 min for 6 hr.

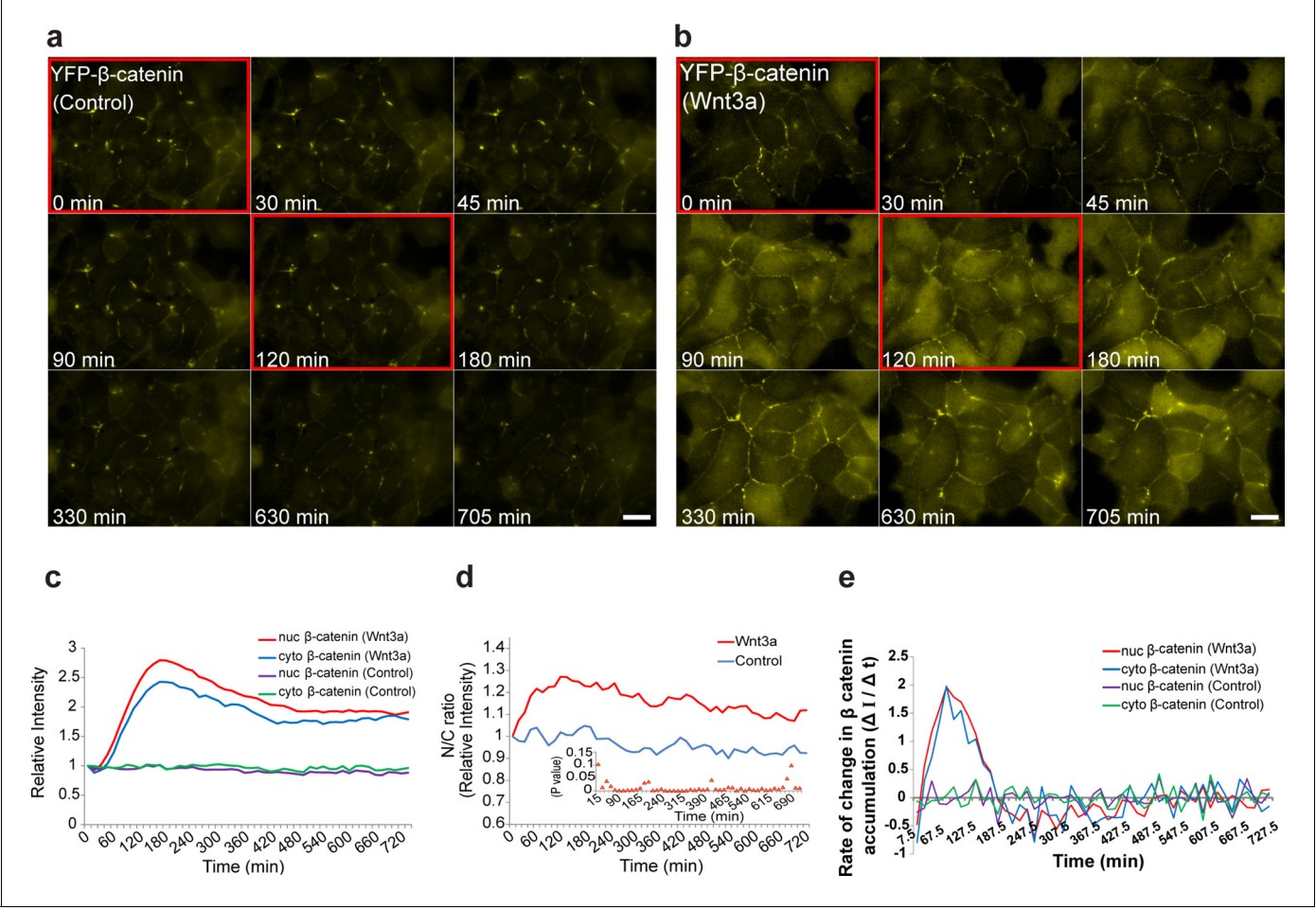

**Figure 2.** The dynamics of β-catenin accumulation following Wnt3a activation in cell populations. Frames from live-cell movies (*Video 2*) showing YFP-β-catenin dynamics in cells treated with (**a**) mock conditioned medium or (**b**) Wnt3a for 12 hr. Red bordered frames compare between the 0 min and 120 min time points. Bar = 20 μm. (**c**) The relative average intensity of β-catenin measured in the cytoplasm (n = 24) and nucleus (n = 31) of cells treated with Wnt3a for 12 hr, compared to mock-treated control cells (n = 13). (**d**) Nucleus to cytoplasm ratio (N/C) of fluorescence intensities over 12 hr from **c**. The initial ratio was designated as 1. Inset plot shows the statistical significance p values (t test) at each time point between the two treatments over the experiment time course. (**e**) The rate of change in β-catenin levels (ΔI/Δt), during accumulation or degradation, in the cytoplasm and nucleus over time in cells from **c**.

The following figure supplement is available for figure 2:

**Figure supplement 1.** FRAP and FLIP measurements of YFP-β-catenin import and export dynamics.

expression conditions (*Figure 2—figure supplement 1a,b*, *Supplementary file 1b*). The incomplete recovery of YFP-β-catenin during the FRAP time-course meant that a significant population of β-catenin molecules had already accumulated and had been retained in the nucleus prior to photobleaching. Next, we photobleached the cytoplasm and found that the rate of β-catenin shuttling out from the nucleus was slower than the import rate (*Figure 2—figure supplement 1a* bottom, c, *Supplementary file 1c*). Similarly, fluorescence loss in photobleaching (FLIP), either in the nucleus or in the cytoplasm, showed that β-catenin shuttling out of the nucleus was slower than its nuclear entry (*Figure 2—figure supplement 1d*, *Supplementary file 1d*). Altogether, the data suggest that Wnt signaling causes a transient shift in the dynamic interplay between β-catenin stabilization and degradation processes, towards protein stabilization and accumulation.

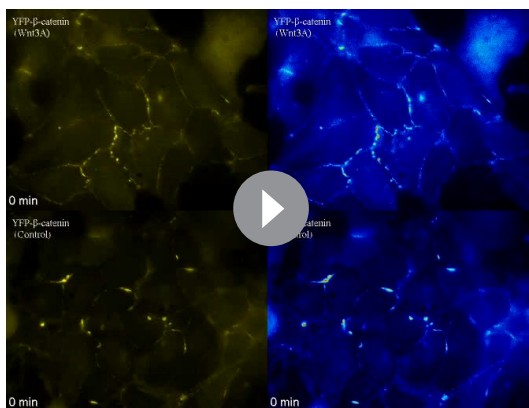

**Video 2.** YFP-β-catenin dynamics at steady state and after Wnt3a activation. HEK293 CCND1-MS2 cells stably expressing YFP-β-catenin were treated with Wnt3a (top) and showed nuclear and cytoplasmic accumulation of YFP-β-catenin, followed by slow egress. No change in YFP-β-catenin levels was seen in mock-treated cells (bottom). Right – The YFP signal is pseudo-colored using ImageJ 'Royal' look-up table to show YFP-β-catenin levels. Cells were imaged every 15 min for 510 min.

## Individual cells in the population present a variable response of β-catenin dynamics

The averaged population data obtained from living cells presented above (**Figure 2**) are in agreement with biochemical data as seen by Western blotting of protein extracts from large cell populations, showing the accumulation of β-catenin beginning from around 30 min after Wnt and peaking at 3 hrs (**Hernández et al., 2012**; **Li et al., 2012**). However, the averaged behavior of a population does not necessarily represent the actual dynamics in individual cells. Examining the dynamic behavior of β-catenin accumulation in the nucleus and cytoplasm of individual cells after Wnt3a showed that although an increase in β-catenin levels was initiated in most cells, the subsequent dynamics were variable (**Figure 3a,b**, **Video 3**). For instance, comparing cells 1,2 and 4 (**Figure 3a**) showed a major and rapid wave of β-catenin nuclear accumulation in cell 1 (30–165 min) that subsided and then mildly rose again (465–585 min); a similar range of events occurred in cell 2 but the two waves were less intense and the second wave occurred earlier compared to cell 1 (first wave 30–150 min, second wave 330–435 min); in contrast, cell four showed a longer accumulation period (30–240 min). Cells 3 and 6 showed slow nuclear accumulation, peaking late only after 825 min and 525 min, respectively, from Wnt3a stimulation. This analysis showed that the dynamic behavior of β-catenin in the cytoplasm and the nucleus was highly similar within the same cell, but that the time-frames of accumulation could be quite different between individual cells, some showing two cycles of nuclear accumulation. In these cases, the first cycle of accumulation lasted 360 min on average and the second cycle 180 min on average.

The similar dynamics of decline in β-catenin levels in the nucleus and the cytoplasm suggests that β-catenin is not simply shuttling in and out of the nucleus, but rather reflects an enhanced activity of the degradation arm controlling β-catenin levels. To test this, we added lithium chloride (LiCl, 20 mM), a glycogen synthase kinase-3β (GSK3β) inhibitor that mimics Wnt signaling (**Klein and Melton, 1996**; **Hedgepeth et al., 1997**). Indeed, LiCl caused β-catenin nuclear and cytoplasmic accumulation, but the dynamics were completely different than Wnt3a (**Figure 3—figure supplement 1a,b**, and **Video 4**). β-catenin accumulation occurred synchronously and continuously throughout 10–11 hr in all cells, and only then did the accumulation cease. The increasing accumulation rate of change (ΔI/Δt) in the nucleus and cytoplasm continued for 10 hr, compared to 3 hr, in response to Wnt3a (**Figure 3—figure supplement 1c,d**). The levels of β-catenin were 4-fold higher in LiCl treated cells compared to Wnt3a. Since LiCl prevents β-catenin degradation, we hypothesized that Wnt3a treatment together with the proteasome inhibitor MG132, which stabilizes β-catenin, but not through GSK3β phosphorylation, should have a similar effect on β-catenin dynamics. Indeed, accumulation dynamics under Wnt3a+MG132 were similar to LiCl treatment (**Figure 3—figure supplement 1e**). Treatment with MG132 without Wnt3a showed the same dynamics (data not shown). When the curve describing the dynamics of β-catenin in response to Wnt3a (**Figure 2c**) was fitted with a two-phase exponential fit that describes production and degradation (**Figure 2—figure supplement 1e**), we found linear accumulation in the first phase, showing that degradation was very low, as expected (**Hernández et al., 2012**). β-catenin production rates did not change significantly during the accumulation and clearance phases, whereas, the degradation rate became predominant during the clearance phase. β-catenin degradation had a characteristic time of 2.75 hr. These data exemplify the difference between a signaling molecule and a chemical that target the same signaling pathway.

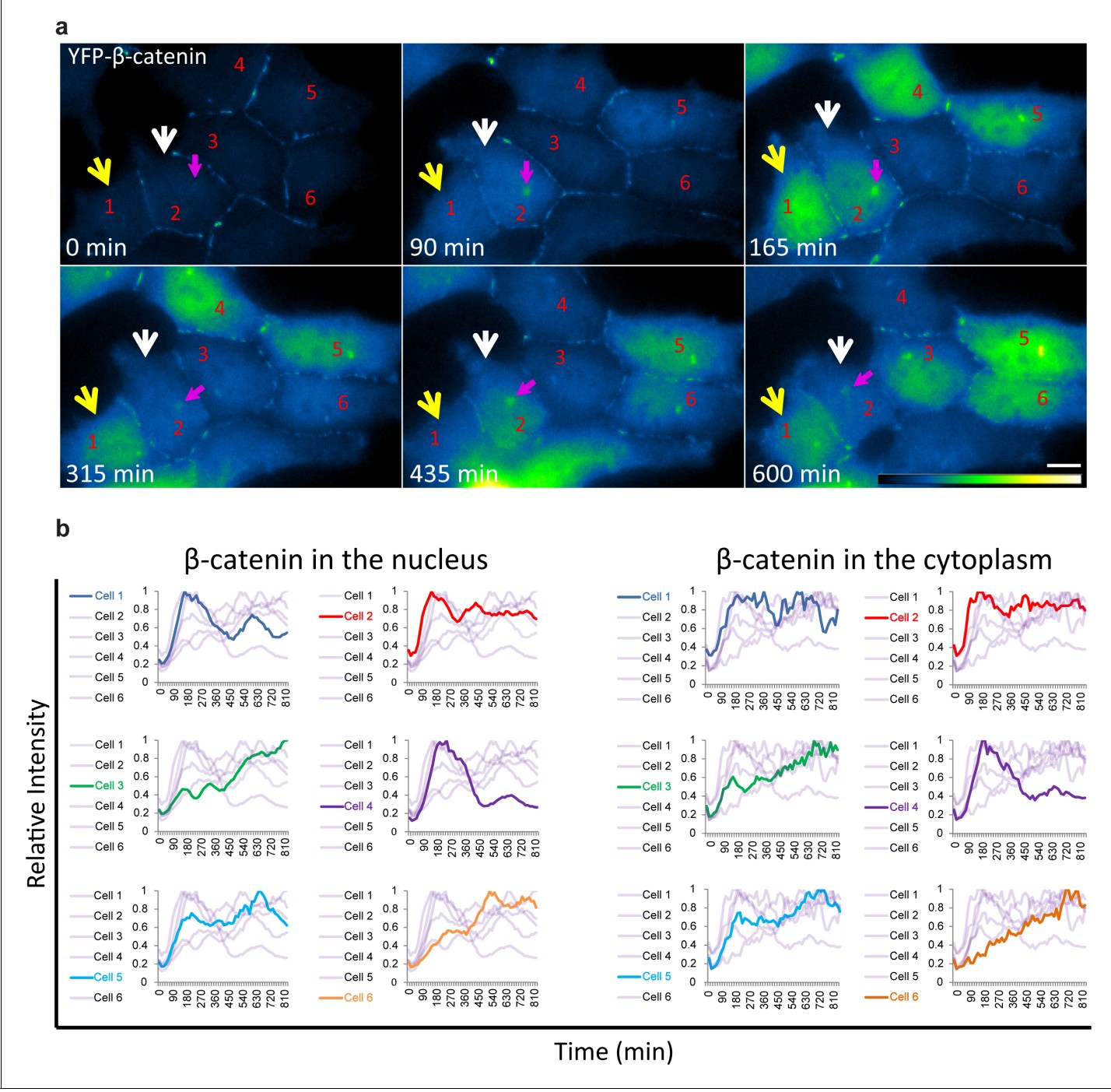

**Figure 3.** Variability of β-catenin accumulation dynamics following Wnt3a activation in individual cells. (**a**) Frames from time-lapse *Video 3* showing YFP-β-catenin accumulation in a population of cells. The YFP signal is pseudo-coloured using ImageJ 'Green Fire Blue' look-up table. White and yellow arrows point to cells in which β-catenin levels increase and decrease twice during the movie. The pink arrow points to centrosomal accumulation. Bar = 10 μm. (**b**) β-catenin levels in the nucleus (left) and cytoplasm (right) in individual cells (as numbered in **a**) are plotted in different colors. The grey background plots show the complete set of plots from all the cells. Maximum β-catenin intensity in each cell was normalized to 1.

The following figure supplements are available for figure 3:

**Figure supplement 1.** β-catenin accumulation dynamics in response to LiCl activation in individual cells.

**Figure supplement 2.** The relationship between YFP-β-catenin levels of accumulation and time of Wnt3a activation.

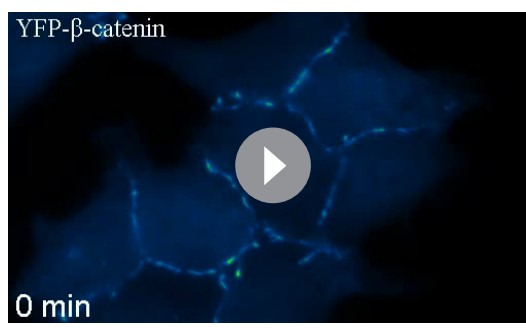

**Video 3.** YFP-β-catenin dynamics in individual cells. HEK293 CCND1-MS2 cells stably expressing YFP-β-catenin were treated with Wnt3a, and the dynamics of the protein were observed in individual cells. The YFP signal is pseudo-colored using ImageJ 'Green Fire Blue' look-up table to show YFP-β-catenin levels. Cells were imaged every 15 min for 825 min.

While drug action is less influenced by endogenous molecules, a signaling molecule will relay a transient signaling effect, depending on the level of other signaling molecules that are present in the cell at the time of induction.

Since the maximum levels of β-catenin accumulation differed between cells in the population (*Figure 3—figure supplement 2a–e*), and we could identify intense and prolonged accumulation in some Wnt3a-treated cells, we examined whether there was a correlation between the time to reach the maximum level and the peak of the response. However, a low correlation score (0.28) was observed for the Wnt3a-treated cells, and a more prominent correlation score (0.53) in LiCl-treated cells (*Figure 3—figure supplement 2f,g*). The latter was expected due to the continuous accumulation over time. But for Wnt3a treatment, this meant that a longer Wnt3a signaling response did not necessarily result in higher levels of β-catenin accumulation. Moreover, calculating the integral of the fluorescence signal that accumulated over the whole observation period in a cell population (from *Figure 3*), showed that the total accumulation in most cells was similar (*Figure 3—figure supplement 2h*), and that differences between single cells were pronounced mainly at earlier time points of the response.

## The response of cells to Wnt3a is not cell cycle dependent

Cluster analysis of the dynamic behavior of β-catenin in individual living cells, shows the dramatic difference between Wnt signaling activation by Wnt3a compared to LiCl (*Figure 4a*; membrane and centrosome will be discussed below). ~80% of the cells showed similar dynamics (e.g. *Figure 2c*) and ~20% portrayed different behavior patterns (e.g. *Figure 3*). In order to determine whether the variabilities in β-catenin dynamics in the cell population in response to Wnt3a, may be due to the cell cycle stage, we examined time-lapse movies in which cells had undergone mitosis, and in which daughter cells could be identified. For example, in the population of cells seen accumulating β-catenin in response to Wnt3a in *Figure 4b* (*Video 5*), there were two dividing cells at the beginning of the movie, both with low β-catenin levels prior to mitosis. In the daughter cells originating from the top dividing cell there was low β-catenin accumulation, whereas in the bottom dividing cell, one daughter cell responded rapidly and accumulated very high levels of β-catenin, while the other daughter cell responded later and accumulated to low levels (*Figure 4b–e*). In summary, we could not detect a pattern of β-catenin accumulation in daughter cells.

To examine the cell cycle and Wnt response more closely in a large population of living cells we used the Fucci system (*Videos 6* and *7*), which uses two fluorescent cell cycle markers to identify cell cycle phases (*Sakaue-Sawano et al., 2008*). We introduced the Fucci molecules into the CCND1-MS2 cells containing YFP-β-catenin. The cells did not show any special pattern of YFP-β-catenin accumulation relative to the cell cycle stage (*Figure 4—figure supplement 1a*), and cells passing through mitosis also

**Video 4.** YFP-β-catenin dynamics in response to LiCl. HEK293 CCND1-MS2 cells stably expressing YFP-β-catenin were treated with LiCl and increased accumulation of the protein was observed. The YFP signal is pseudo-colored using ImageJ 'Green Fire Blue' look-up table to show YFP-β-catenin levels. Cells were imaged every 15 min for 825 min.

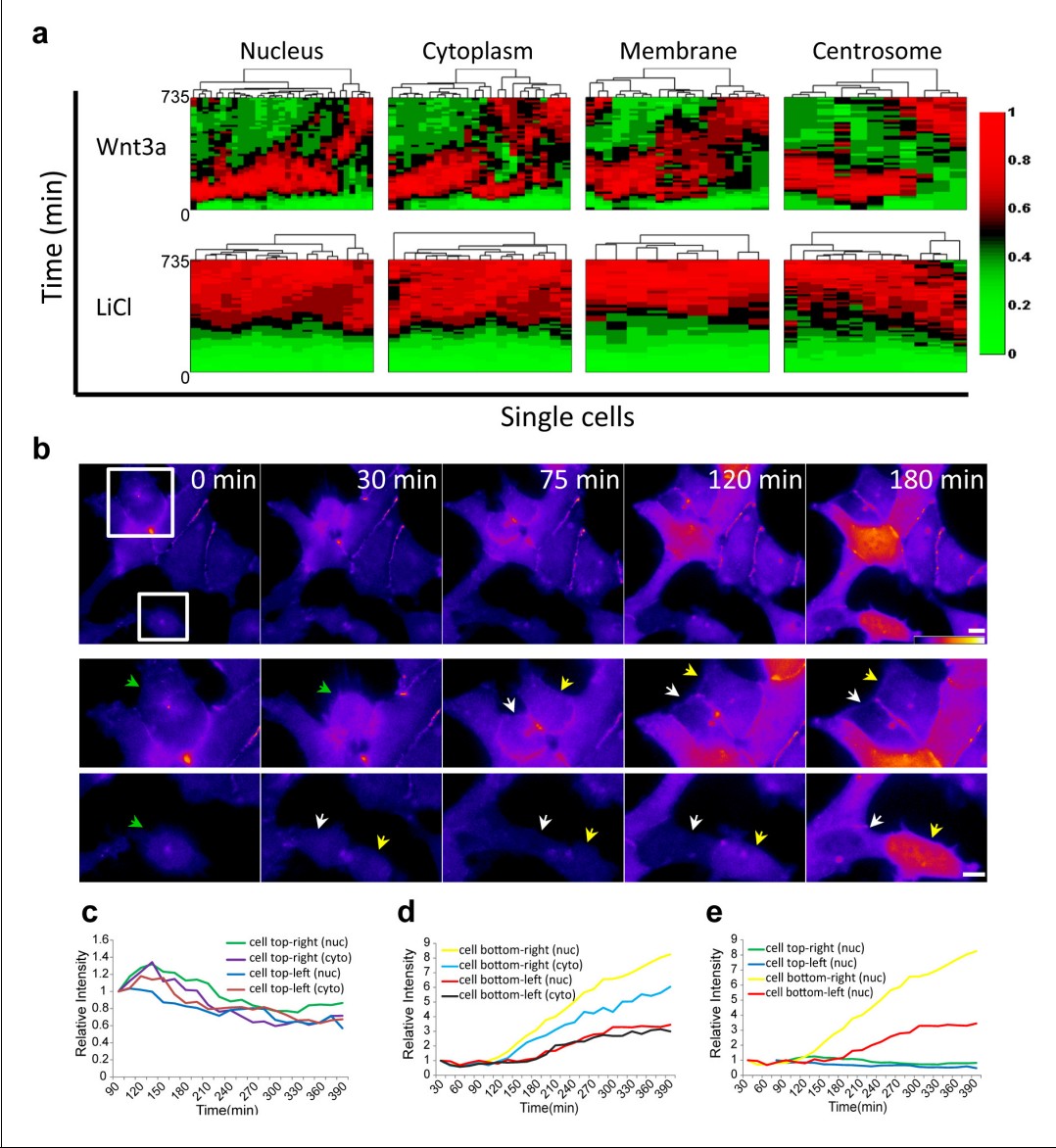

**Figure 4.** Variability of β-catenin dynamics in the cell population and during the cell cycle. (a) Heat map and cluster analysis of normalized β-catenin accumulation dynamics in sub-cellular compartments following Wnt3a (top, n(nucleus) = 31, n(cytoplasm) = 24, n(membrane) = 21, n(centrosome) = 11) or LiCl (bottom, n(nucleus) = 18, n(cytoplasm) = 17, n(membrane) = 9, n(centrosome) = 14) treatments. Data were taken from live-cell movies with each column representing one cell, and rows representing time from Wnt addition. Relative levels of β-catenin are depicted from low (green) to high (red). Hierarchical cluster analysis depicted above the plots shows the homogenous behavior in LiCl-treated cells and heterogenous behavior in Wnt3a-treated cells. Most cells reach maximal levels of β-catenin within 2–3 hr. (b) (Top) Frames from time-lapse *Video 5* showing YFP-β-catenin accumulation in a population of cells. The YFP signal is pseudo-colored using the ImageJ 'Fire' look-up table. Boxes denote cells that go through mitosis, and enlargements are shown below. Green arrows point to mother cells, and yellow and white arrows point to the daughter cells. Bar = 10 μm. Plots showing the relative intensity levels of YFP-β-catenin in the cytoplasm and nucleus of the (c) top and (d) bottom daughter cells of each cell division. (e) Plot comparing the relative intensity levels in the nuclei of the four daughter cells.

The following figure supplement is available for figure 4:

**Figure supplement 1.** YFP-β-catenin dynamics during the cell cycle in Wnt3a induced cells.

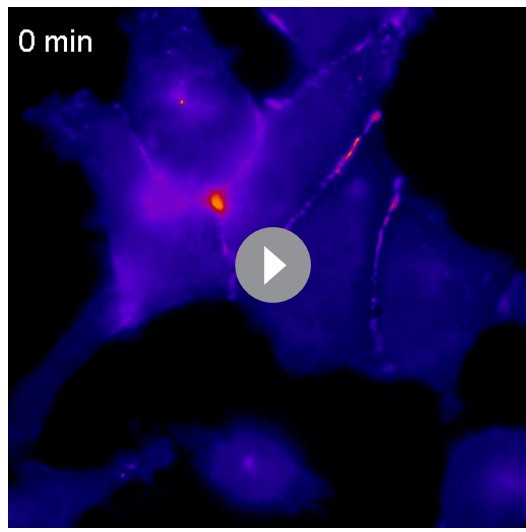

**Video 5.** YFP-β-catenin dynamics following Wnt3a activation during cell division. HEK293 CCND1-MS2 cells stably expressing YFP-β-catenin were treated with Wnt3a, and the dynamics of the protein in the nucleus were followed over time. Two cells that undergo mitosis were observed in the field. The levels of the protein in the daughter cells formed from the upper cell were low (also note the appearance and division of the centrosome detected via YFP-β-catenin). In comparison, in the bottom mitotic cell, one daughter cell accumulated high YFP-β-catenin levels very rapidly, while the other responded slowly and had very low levels. The YFP signal is pseudo-colored using ImageJ 'Fire' look-up table to show YFP-β-catenin levels. Cells were imaged every 15 min for 225 min.

exhibited different accumulation levels in the mother cell and between daughter cells (*Figure 4—figure supplement 1b*). In summary, we did not identify a cell cycle dependent pattern of YFP-β-catenin levels in response to Wnt.

## Wnt signaling induces β-catenin accumulation at the cell membrane and the centrosome

β-catenin is normally present in the adherens junctions proximal to the cell membrane, and is bound to E-cadherin in the membrane and to α-catenin, which mediates the connection between the adherens junction and the actin cytoskeleton (*Yap et al., 1997*; *Brembeck et al., 2006*). Not much is known about the subcellular localization of this β-catenin population in response to Wnt. Before treatment, β-catenin was observed as a string of punctate sub-regions distributed along the cell outline only at cell-cell contacts (*Figure 3a*, *Video 3*). Since we could detect changes in the intensity of the puncta after Wnt, we followed the intensity of β-catenin at the membrane during Wnt activation and found an increase with similar dynamics to the cytoplasmic and nuclear sub-populations (*Figure 5a*, *Video 8*). There was no obvious reduction in the membrane levels even after many hours (*Figure 5b*). However, the relative increase at the membrane was lower than the nucleus and the cytoplasm, and the rate of β-catenin accumulation ($\Delta I/\Delta t$) at the membrane was less rapid than the nuclear accumulation rates (*Figure 5b,c*). LiCl caused longer β-catenin accumulation times and significantly higher accumulation at the membrane (*Figure 5c,d*).

To examine if Wnt signaling changed the dynamics of β-catenin at the membrane we performed FRAP experiments on this region and found that the recovery dynamics were slow and indicative of slow exchange of β-catenin molecules at the membrane. Yet, similar recovery in unactivated, Wnt3a-treated and LiCl-treated cells was observed, meaning that there was no change in the dynamics of protein exchange but rather an increase in the

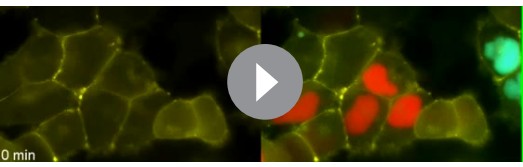

**Video 6.** YFP-β-catenin dynamics following Wnt3a activation during the cell cycle. HEK293 CCND1-MS2 cells stably expressing YFP-β-catenin (yellow) and the Fucci markers for G1 (red) and G2 (cyan), were treated with Wnt3a, and the dynamics of the protein were followed over time. Cells were imaged every 15 min for 1065 min.

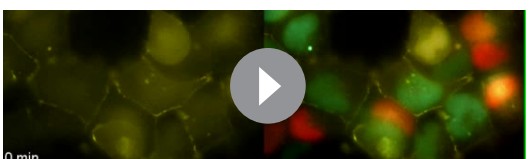

**Video 7.** YFP-β-catenin dynamics following Wnt3a activation during cell division. HEK293 CCND1-MS2 cells stably expressing YFP-β-catenin (yellow) and the Fucci markers for G1 (red) and G2 (cyan), were treated with Wnt3a, and the dynamics of the protein were followed over time in four cells that undergo mitosis. Cells were imaged every 15 min for 705 min.

number of β-catenin molecules in the membrane-bound fraction (*Figure 5—figure supplement 1*, *Supplementary file 1e,f*).

In many of the Wnt-induced cells that were followed in the live-cell movies we noticed the appearance of β-catenin in a single prominent dot (*Figure 3a*, *Video 3*). β-catenin can localize at the centrosome during interphase and mitosis, and functions in centriolar cohesion (*Kaplan et al., 2004*; *Hadjihannas et al., 2010*; *Bahmanyar et al., 2008*). Since the β-catenin dot was in proximity to the nucleus, and since the centrosome is juxtaposed to the nucleus, we examined if centrosomal accumulation of β-catenin was occurring. Indeed, movies of dividing cells demonstrated that each daughter cell received one β-catenin-labeled body after division, reminiscent of centrosome behavior (*Figure 6a*, *Video 5* and *9*). Immunofluorescence of pericentrin (a centrosome marker), together with either endogenous β-catenin or YFP-β-catenin, showed accumulation of β-catenin at the centrosome following activation (*Figure 6b*).

The accumulation dynamics of β-catenin at the centrosome occurred in parallel to the accumulation seen in the nucleus, cytoplasm and adherens junctions. However, centrosomal levels were significantly higher, five-fold higher compared to the initial state (*Figure 6c*). The rates of change were the highest and most rapid of all measured cell compartments (*Figure 6d*). LiCl also led to β-catenin localization at the centrosome, but here too with very different dynamics from Wnt3a (*Figure 6d*; *Figure 3—figure supplement 1f*). To obtain a more general outlook of the changes in β-catenin levels in all four compartments, we performed a correlation analysis (*Figure 6c,e*). As was seen in individual cells, the highest correlation in accumulation dynamics following Wnt3a, was observed between the cytoplasm and the nucleus, whereas the lowest correlation was between the centrosome and the membrane.

Interestingly, in some cells we observed β-catenin puncta detaching from the membrane and traveling in the cell (*Videos 10* and *11*). When these structures were tracked during movement in the cell, they usually ended up at the centrosome (*Figure 6—figure supplement 1*). This phenomenon was frequently seen in cells treated with Wnt3a, LiCl and MG132, and less frequently in unactivated cells. We did not observe a correlation with the timing of Wnt addition, and perhaps detection was easier after Wnt due to the increase of β-catenin at the membrane following stimulation. Tracking of the detached β-catenin puncta showed that they reached the centrosome between 30 to 90 min after detachment. To examine whether the residence times of β-catenin molecules at the centrosome resembled the membrane region, we performed FRAP analysis, which showed very rapid recovery kinetics at the centrosome, in comparison to all other cell regions (*Figure 6—figure supplement 2*). This implied that β-catenin duration at the centrosome is short-lived, with a half-time of fluorescence recovery ($t_{1/2}$) of 1.9 s, similar to other centrosomal components (*Hames et al., 2005*). Altogether, this suggests that the molecular interactions of β-catenin at the membrane in adherens junctions are significantly more stable than at the centrosome, where the exchange of β-catenin molecules is highly rapid.

## Wnt signaling modulates the transcriptional output of the *cyclin D1* gene

We next examined the influence of Wnt signaling dynamics on *CCND1* gene activity. As shown (*Figure 1a*), a significant increase in the percentage of cells actively transcribing CCND1-MS2 could be seen starting 15 min post-activation, and peaking after 75–90 min. Cells returned to steady state activity levels after 6 hr. We examined several parameters of the transcriptional response. First, we measured the time for an active CCND1-MS2 transcribing gene to appear in the population. In the control unstimulated population (mock conditioned medium), after 120 min most cells had activated the gene once, whereas in Wnt3a-induced cells, gene activation in the population was reached more quickly, already after 60 min. The response time for *CCND1* activation following Wnt3a was also short, ranging at 15 min (*Figure 7a–b*). This meant that Wnt signaling increased the probability of *CCND1* to initiate transcription.

We next examined whether the periods of gene activity were altered after Wnt activation. When *CCND1* was at first non-active and began to transcribe after Wnt3a, there was prolonged transcriptional activation for a time-frame of 180 min, compared to a shorter activity period of 65 min in unactivated cells (*Figure 7c,e*, *Video 12*). This meant that Wnt signaling increased the time-frame of *CCND1* promoter activity. Surprisingly, if the gene was detected in an already active state, and Wnt3a was then added, there was no difference in the activity period compared to that in

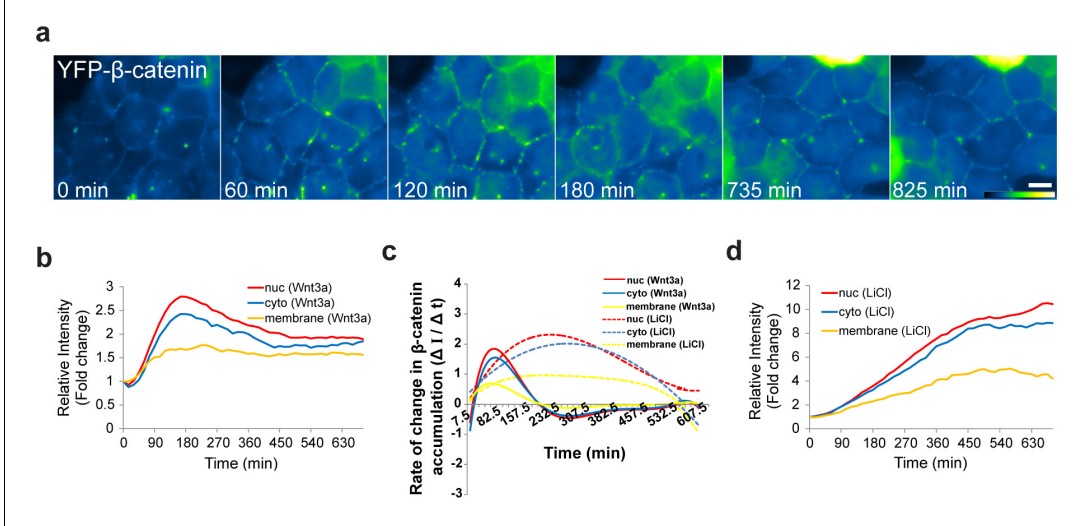

**Figure 5.** The dynamics of β-catenin accumulation at the membrane following Wnt3a activation. (a) Frames from time-lapse *Video 8* showing YFP-β-catenin accumulation at the cell membrane. The YFP signal is pseudo-colored using the ImageJ 'Green Fire Blue' look-up table. Bar = 10 μm. (b) The relative average intensity of β-catenin measured in the membrane (n = 21), cytoplasm and nucleus (from *Figure 2*) of Wnt3a-treated cells. (c) The rate of change in β-catenin levels (ΔI/Δt) accumulation or degradation in the membrane, cytoplasm and nucleus over time in Wnt3a- and LiCl-treated cells. (d) The relative average intensity of β-catenin measured in the membrane, cytoplasm and nucleus of LiCl-treated cells.

The following figure supplement is available for figure 5:

**Figure supplement 1.** FRAP measurements of YFP-β-catenin dynamics at adherens junctions.

unactivated cells. Under both conditions, activation persisted for an average of 130 min (*Figure 7d*), meaning that if the promoter was already activated then there was no Wnt-induced change in this time-frame.

When we examined the levels of CCND1-MS2 activity after Wnt activation in living cells (*Figure 8—figure supplement 1a–c*), we found that even if the gene was active before Wnt3a addition, the intensity of MS2-GFP fluorescence on the gene showed higher levels, indicative of higher expression levels due to signaling, meaning that the promoter could integrate additional signals (*Figure 8—figure supplement 1b*). We measured a 1.5–1.7 increase in the maximum MS2-GFP intensity levels, and observed that the maximum intensity distribution for Wnt3a-treated cells shifted such that many more cells displayed higher levels of gene activity (*Figure 8a,b*, and *Figure 8—figure supplement 1d*). The time required to reach the maximum point of activity did not seem to change when examining the whole population (*Figure 8—figure supplement 1e*). However, this time was actually shortened from 170 min to 120 min in cells where the gene was initially inactive, and the distribution of cells shifted to shorter times to reach maximum levels of transcription

**Video 8.** YFP-β-catenin dynamics at the cell membrane following Wnt3a activation. HEK293 CCND1-MS2 cells stably expressing YFP-β-catenin were treated with Wnt3a, and the dynamics of the protein at the membrane were followed over time, and were similar to the nucleus and cytoplasm accumulation. The YFP signal is pseudo-colored using ImageJ 'Green Fire Blue' look-up table to show YFP-β-catenin levels. Cells were imaged every 15 min for 1065 min.

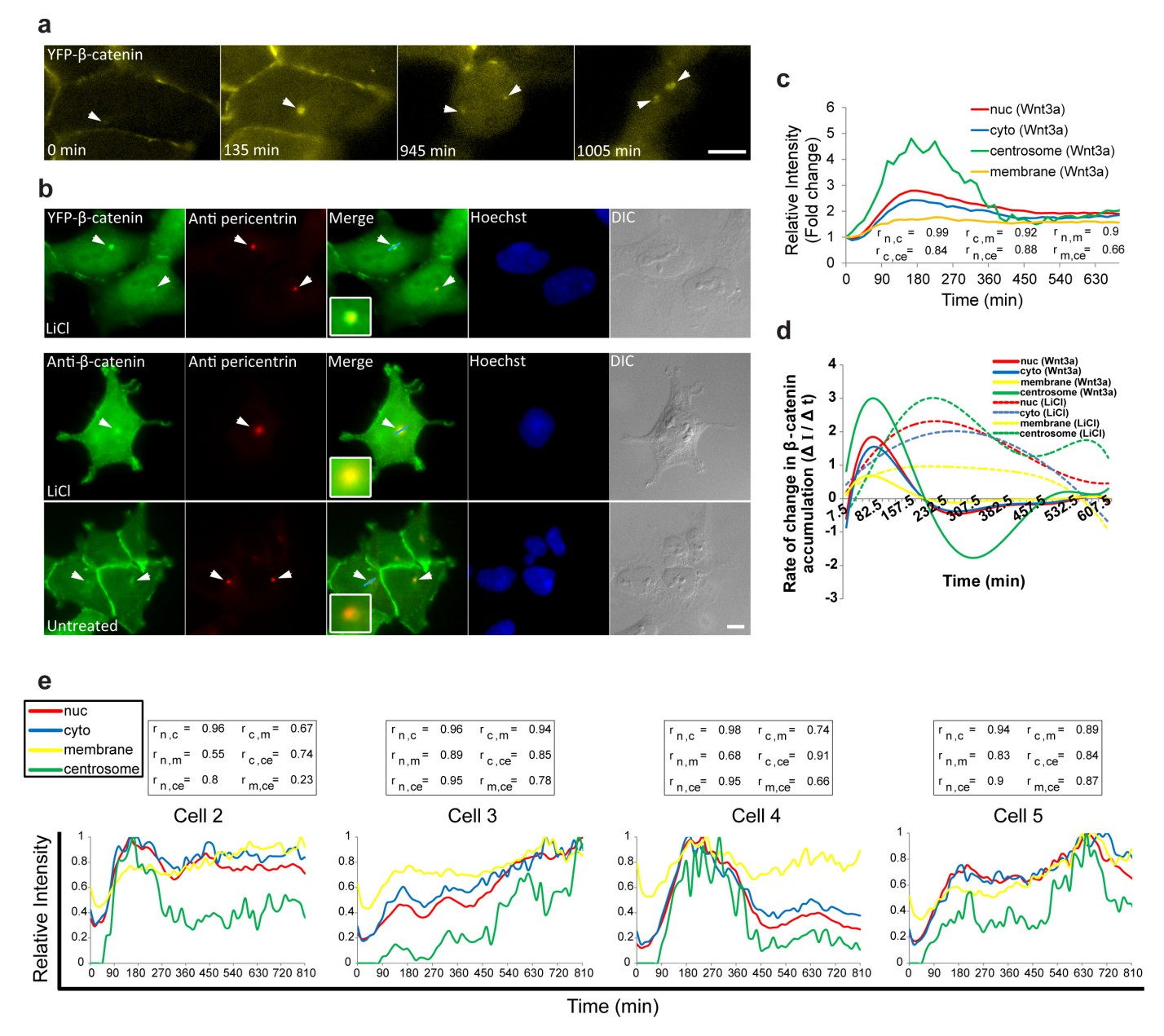

**Figure 6.** Accumulation of β-catenin at the centrosome after Wnt3a activation. (**a**) Frames from time-lapse *Video 9* showing YFP-β-catenin accumulation at the centrosome (white arrowheads) and after cell division. Bar = 10 μm. (**b**) The colocalization (white arrowheads) of YFP-β-catenin (top) or endogenous β-catenin (bottom) with the centrosomal marker pericentrin (red immunofluorescence) in untreated and LiCl-treated cells. Hoechst DNA stain is in blue, and DIC in grey. Boxes show enlarged centrosomal areas. Bar = 10 μm. (**c**) The relative average intensity of YFP-β-catenin measured in the centrosome (n = 11), membrane, cytoplasm and nucleus (from *Figure 2 and 5*) of Wnt3a-treated cells. Correlation scores (r) between the nucleus (n), cytoplasm (c), membrane (m) and centrosome (ce) YFP-β-catenin levels are presented at the bottom. (**d**) The rate of change in YFP-β-catenin levels (ΔI/Δt) accumulation or degradation in the centrosome, membrane, cytoplasm and nucleus over time in Wnt3a- and LiCl-treated cells. (**e**) Plots of YFP-β-catenin levels in the sub-cellular compartments of individual cells (from *Figure 3*). Boxes show the correlation scores (r) between the nucleus (n), cytoplasm (c), membrane (m) and centrosome (ce).

The following figure supplements are available for figure 6:

**Figure supplement 1.** Detachment of membranal YFP-β-catenin puncta and movement towards the centrosome.

**Figure supplement 2.** Summary of FRAP measurements of YFP-β-catenin dynamics in subcellular compartments in response to Wnt3a treatment.

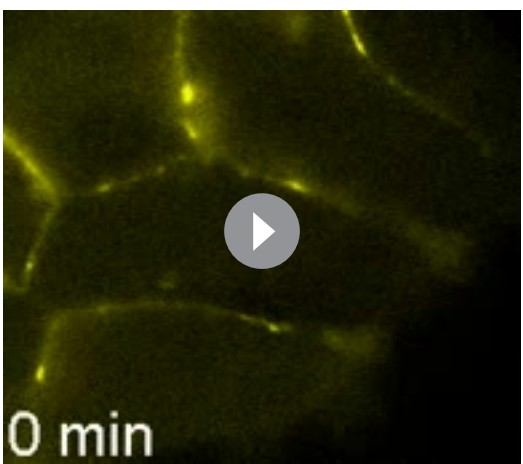

**Video 9.** YFP-β-catenin accumulation at the centrosome following Wnt3a activation. HEK293 CCND1-MS2 cells stably expressing YFP-β-catenin were treated with Wnt3a, and the dynamics of the protein at the centrosome were observed in parallel to the accumulation in the nucleus and cytoplasm. The separation of the centrosome in a cell during division can be seen after the 960 time point. Cells were imaged every 15 min for 1005 min.

(*Figure 8c*). This time did not change in cells where the gene was initially active (*Figure 8d*). When gene activity and gene inactivity patterns were further examined, not only was an expected increase in the duration of gene activity found, but also a reduction in the rest duration. This means that Wnt activation not only increases the duration time for gene activity, but also reduces periods of inactivity by increasing the frequency of promoter firing events (*Figure 7—figure supplement 1*).

These measurements suggested that Wnt3a signaling increases promoter firing events so that more CCND1 mRNAs are transcribed. To further examine this on the single mRNA level, we performed quantitative RNA FISH on CCND1-MS2 mRNA molecules in parallel to measuring β-catenin nuclear levels within the same single cell (fixed cells). We counted the number of cellular and nascent CCND1-MS2 mRNAs in Wnt3a-treated cells (*Figure 8e*) and compared this value to the accumulation levels of nuclear β-catenin in the different cells. Cells that had accumulated β-catenin had significantly higher numbers of cellular CCND1-MS2 mRNAs (3-fold; *Figure 8e,f*) and nascent CCND1-MS2 mRNAs (3.8-fold; *Figure 8e,g,j*), which correlated well with the

transcription measurements in living cells (*Figure 8—figure supplement 1c*). Correlating between cellular and nascent CCND1-MS2 mRNA

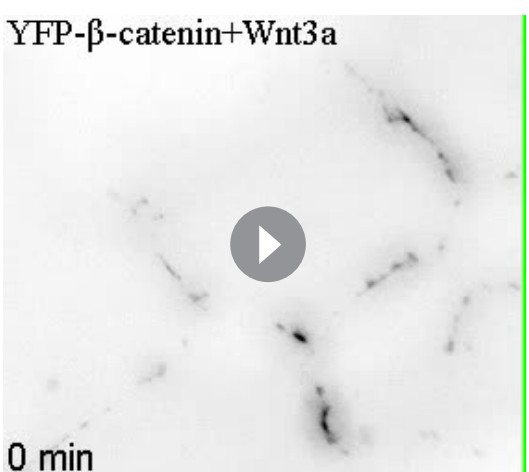

**Video 10.** YFP-β-catenin puncta move from the membrane to the centrosome. HEK293 CCND1-MS2 cells stably expressing YFP-β-catenin were treated with Wnt3a. At the 300 min time point, a series of YFP-β-catenin puncta can be tracked (track colors) moving from the membrane to the centrosome. An inverted presentation of the movie shows the movie puncta (black dots). Cell was imaged every 15 min for 1005 min.

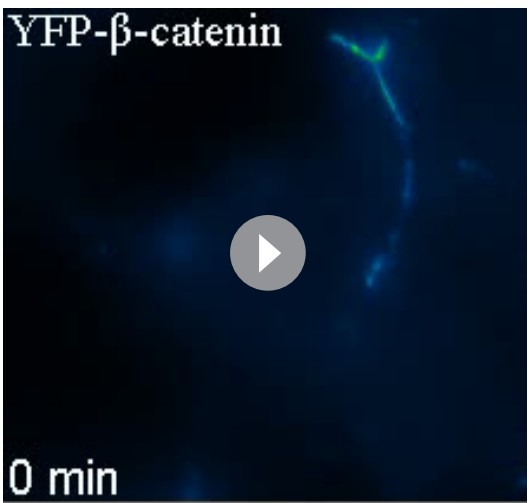

**Video 11.** YFP-β-catenin puncta move from the membrane to the centrosome. HEK293 CCND1-MS2 cells stably expressing YFP-β-catenin were treated with Wnt3a and MG132. At the 165 min time point, a series of YFP-β-catenin puncta can be tracked (track colors) moving from the membrane to the centrosome. The YFP signal is pseudo-colored using ImageJ 'Green Fire Blue' look-up table to show YFP-β-catenin levels. Cells were imaged every 15 min for 1065 min.

numbers and β-catenin levels showed two sub-populations of high- and low-expressing CCND1-MS2 cells, in correlation with nuclear β-catenin accumulation, respectively (*Figure 8h,i*). Regarding gene activation, altogether we find that Wnt signaling leads to increased promoter firing frequency, increased gene activity duration time, reduced gene rest time, and significantly higher numbers of mRNAs in the cell.

## Discussion

Signaling factors that translocate into the nucleus following signal transduction do so via different modes of shuttling. For instance, some factors display continuous nucleo-cytoplasmic oscillations (p53, mdm2, NF-κB, ERK) (*Purvis and Lahav, 2013*; *Kalo and Shav-Tal, 2013*; *Lahav et al., 2004*; *Shankaran et al., 2009*), while some show a rapid and limited pulse of nuclear build-up (NFAT) (*Yissachar et al., 2013*), or a prolonged presence in the nucleus (MAL) (*Cui et al., 2015*). These dynamics have been characterized using microscopy studies performed in single cells. Biochemical examination of these dynamics can give a true sense of the time-scales of the accumulation as seen by studying protein levels in Western blots (*Hernández et al., 2012*; *Lustig et al., 2002*). However, such approaches cannot provide an accurate temporal dynamic profile of the response as it unfolds within the cell, since they represent an average picture of the behavior of the whole cell population from which the proteins were extracted (*Levsky and Singer, 2003*). By characterizing β-catenin accumulation dynamics in several subcellular compartments within individual living cells, we could examine how varying responses in individual cells translate into a well-timed response of the cell population.

Using a cell system we previously generated to follow CCND1 transcription in real-time on the single gene level (*Yunger et al., 2010*), we now measured β-catenin sub-cellular dynamics, as well as characterized the transcriptional response of *CCND1* to Wnt. Even though Wnt/β-catenin signaling has been highly studied, the basic propagation dynamics of this signal in single living cells have not been characterized. This is due to the lack of an appropriate system that would allow analysis of the behavior of a fluorescent β-catenin protein that resembles the endogenous protein (*Tan et al., 2014*). Previous studies using transiently overexpressed β-catenin and photobleaching methods were important in establishing the characteristics of its intra-cellular mobility (*Krieghoff et al., 2006*; *Jamieson et al., 2011*). However, the subcellular distribution of transiently overexpressed fluorescent β-catenin is different than the endogenous protein, since the overexpressed protein is found throughout the whole cell including the nucleus (even without a signal), membrane staining is lacking, and cytoskeletal organization is disrupted (*Krieghoff et al., 2006*; *Jamieson et al., 2011*; *Ligon et al., 2001*). Even the use of nanobodies targeting endogenous β-catenin in living cells did not mimic the membrane localization of non-induced cells (*Traenkle et al., 2015*). Hence, using our cell system in which YFP-β-catenin was stably expressed at relatively low levels (80% over the endogenous protein) and was distributed similarly to the endogenous protein, we were able to follow the subcellular dynamics of β-catenin in real-time.

Upon Wnt activation, β-catenin levels in the cell population portrayed a relatively rapid increase in the cells. The general time-scale of hours of β-catenin accumulation concurred with Western blotting experiments (*Hernández et al., 2012*; *Lustig et al., 2002*), and altogether portrayed an orchestrated response of the cell population to the Wnt signal. However, examination of the accumulation profiles in single cells showed response patterns deviating from the average behavior in at least 20% of the population; accumulation rates and levels varied, and in some cells additional but less intense waves of β-catenin nuclear accumulation were observed. We suggest that the balance between accumulation and degradation affects the outcome in β-catenin build-up in each cell. The Kirschner group has shown (*Hernández et al., 2012*) that Wnt does not completely abolish the activity of the destruction complex. We therefore suggest that if the total levels of accumulation are similar in most cells (e.g. integral analysis), then the level of inhibition of the destruction complex is expected to vary in each cell and to determine the response.

However, the fact that *CCND1* transcriptional activation occurs within the same time frame as the main initial phase of β-catenin nuclear accumulation means that in most cells in the population, the *CCND1* gene will become activated shortly after Wnt activation. Possibly, later phases of β-catenin nuclear accumulation could have an influence on prolonging *CCND1* activity (steady state activity levels return after 6 hr). Indeed, measurements of *CCND1* activity in living cells following Wnt

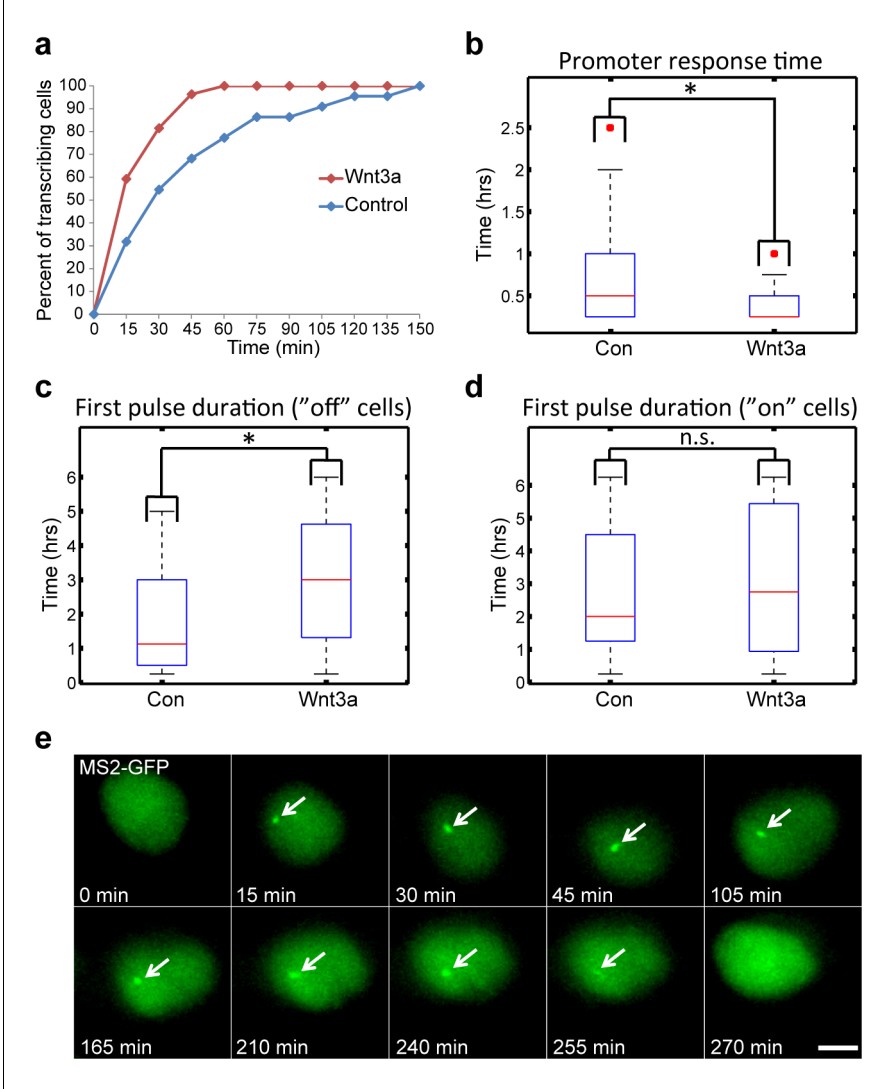

**Figure 7.** Measuring the transcriptional response of *CCND1-MS2* to Wnt3a activation in living cells. (a) The percentage of cells in a population of either mock-treated (blue) or Wnt3a-activated cells (red) showing an active CCND1-MS2 transcribing gene, over time. (b) The promoter response time of *CCND1-MS2* activation from the addition of Wnt3a (n = 27) or in mock-treated conditions (n = 22). In the boxplots, the median is indicated by a red line, the box represents the interquartile range, the whiskers represent the maximum and minimum values, and red dots represent outliers. (p=0.01). (c,d) Periods of gene activity measured in mock-treated and Wnt3a-treated cells. The population was divided into cases where the gene was either not transcribing before the addition of Wnt3a or mock-treatment ('off', n(Wnt3a) = 27, n(Con) = 22, p=0.01) or if the gene was already active ('on', n (Wnt3a) = 37, n(Con) = 52, p=0.77). *p<0.05, n.s. = p>0.05. (e) Frames from *Video 12* showing the activation of the *CCND1-MS2* gene detected by MS2-GFP mRNA tagging (arrow) following Wnt3a treatment. Bar = 10 µm.

The following figure supplement is available for figure 7:

**Figure supplement 1.** Wnt signaling causes shorter rest duration in addition to an increase in the gene burst duration.

activation showed a positive change in several parameters relating to gene activation; not only did the frequency of *CCND1* activation in the cell population rise and the time to activate *CCND1* shorten, but the levels of *CCND1* transcriptional output increased, the timeframe of gene activity became substantially longer, and gene resting periods were shortened. Overall, this means that Wnt signaling increases the number of CCND1 mRNAs generated, by increasing the frequency of RNA

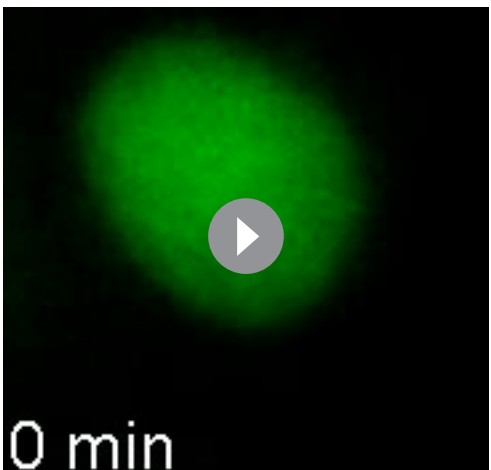

**Video 12.** Prolonged activation of CCND1 after Wnt activation. HEK293 CCND1-MS2 cells stably expressing MS2-GFP (green) were treated with Wnt3a. CCND1 mRNA transcription could be detected 15 min after Wnt3a (green dot, transcription site) and continued for 4 hr. Cells were imaged every 15 min for 270 min.

polymerase II recruitment to the promoter and by lengthening the time of promoter responsiveness. Interestingly, even when an already active *CCND1* gene received the Wnt signal, the levels of gene activity increased.

Although we were unable to examine YFP-β-catenin dynamics and CCND1-MS2 transcription activity simultaneously in living cells, by integrating the measurements of *CCND1* transcriptional activity with the measured dynamics of β-catenin nuclear accumulation from the separate experiments, we found that the rate of change of nuclear β-catenin correlated well with transcription induction (*Figure 9a*), specifically during the first rapid phase of nuclear β-catenin accumulation. This fits in well with a previous study demonstrating that the fold change in β-catenin nuclear levels is the element affecting target gene activity (*Goentoro and Kirschner, 2009*), and that the transcriptional machinery is capable of computing the fold change in β-catenin, thereby determining the transcriptional response (*Goentoro and Kirschner, 2009*). Similar behavior was observed for the ERK signaling pathway (*Cohen-Saidon et al., 2009*). Hence, it is not the absolute number of β-catenin molecules in the nucleus that correlates with transcription rates, but the rate of change of β-catenin levels over time, and particularly the rapid change during the first phase of induction that elicits the transcriptional effect (*Figure 9b*). The advantage of such a sensing mechanism would be its ability to buffer out cellular noise and variability in the cell population.

Concurrent β-catenin accumulation the cell membrane and the centrosome were quantified. β-catenin demarcates the cell outline when there are cell-cell contacts due to its presence in adherens junctions (*Harris and Tepass, 2010*). Generally, while the nuclear accumulation of β-catenin has been the focus of Wnt signaling studies, the membrane region has not been considered a major target of the response. However, one study has shown localization of unphosphorylated β-catenin to the membrane upon Wnt activation within 30 min, in cells lacking E-cadherin, although the function was unclear (*Hendriksen et al., 2008*). We found increased β-catenin levels in the membrane following Wnt activation. The punctate membranal pattern persisted during activation, suggesting that Wnt increases the recruitment of β-catenin to existing adherens junctions. Indeed, β-catenin dynamics in the membrane showed a relatively slow exchange both before and after Wnt activation, similar to a study conducted in LiCl induced cells (*Johnson et al., 2009*). This implies long residence times of β-catenin in the membrane and that potential binding sites for β-catenin molecules at adherens junctions exist constantly, and only when the protein becomes abundant, do they fill up.

Centrosomal localization of β-catenin is known (*Kaplan et al., 2004*; *Hadjihannas et al., 2010*; *Bahmanyar et al., 2008, 2010*; *Mbom et al., 2014*; *Huang et al., 2007*; *Vora and Phillips, 2015*). The exact function is not clear and it probably plays a role in regulation of cell separation. It has been suggested that Wnt signaling abolishes the phosphorylation of β-catenin and leads to centrosome splitting (*Hadjihannas et al., 2010*). Our study shows for the first time, the highly rapid accumulation rates of β-catenin at the centrosome in real-time, following Wnt signaling. β-catenin at the centrosome is highly mobile as seen in our FRAP study and in another (*Bahmanyar et al., 2008*). Interestingly, we identified a connection between the membranal and centrosomal β-catenin fractions. Puncta of membranal β-catenin were detected moving relatively slowly from the membrane region and ending up at the centrosome, sometimes several in parallel in the same cell. Since unphosphorylated β-catenin is found in the membrane after Wnt (*Hendriksen et al., 2008*), we can postulate that the β-catenin fraction moving to the centrosome is unphosporylated, and may be involved in driving cell division.

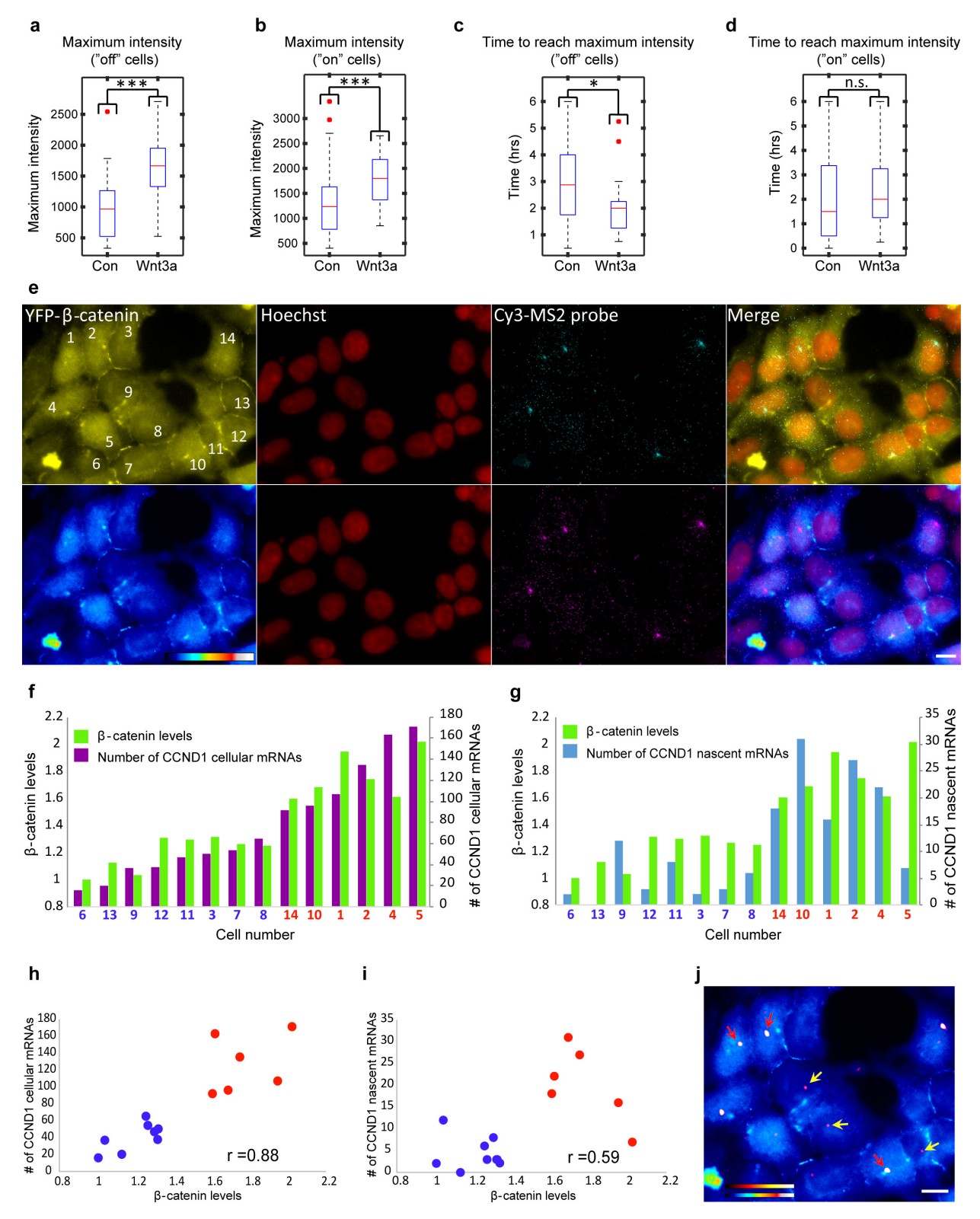

**Figure 8.** Quantification of CCND1 activity levels following Wnt activation in single fixed and living cells. (**a,b**) Boxplots showing the maximal MS2-GFP intensity levels reached on actively transcribing CCND1-MS2 genes during 6 hr in Wnt3a-treated and mock-treated (Con) cells, when (**a**) the gene was either not transcribing before the addition of Wnt3a or mock-treatment ('off', n(Wnt3a) = 27, n(Con) = 22, p=0.0001) or (**b**) if the gene was already active ('on', n(Wnt3a) = 37, n(Con) = 52, p=0.0006). The median is indicated by a red line, the box represents the interquartile range, the whiskers represent

*Figure 8 continued on next page*

*Figure 8 continued*

the maximum and minimum values, and red dots represent outliers. (**c,d**) Boxplots showing the time required to reach the maximal intensity levels when (**c**) the gene was either not transcribing before the addition of Wnt3a ('off', p=0.03) or (**d**) if the gene was already active ('on', p=0.42). (**e**) YFP-β-catenin (yellow) together with RNA FISH images obtained with a probe hybridizing to the MS2 region in the 3'UTR of the CCND1-MS2 mRNA (cyan), showing CCND1 nascent mRNAs on active genes (large dots) and cellular mRNAs (small dots) in Wnt3a-treated cells (2 hr), in comparison to YFP-β-catenin levels. Nuclei are stained with Hoechst (pseudo-colored red). Bottom row is the pseudo-colored YFP signal using the ImageJ 'Royal' look-up table. Cells are numbered. Bar = 10 μm. (**f**) Quantification of the number of cellular CCND1-MS2 mRNAs (ordered from low to high) compared to YFP-β-catenin levels. (**g**) Quantification of the number of nascent CCND1-MS2 mRNAs compared to YFP-β-catenin levels. (**h,i**) Correlation analysis between (**h**) the number of cellular CCND1-MS2 mRNAs and YFP-β-catenin levels and (**i**) between the number of nascent CCND1-MS2 mRNAs and YFP-β-catenin levels. Blue dots – subpopulation with low nuclear YFP-β-catenin levels and low numbers of cellular/nascent CCND1-MS2 mRNAs. Red dots – subpopulation with high nuclear YFP-β-catenin levels and high numbers of cellular/nascent CCND1-MS2 mRNAs. Total correlation score between the number of cellular/nascent CCND1-MS2 mRNAs and YFP-β-catenin levels is 0.88 and 0.59, respectively. (**j**) The field from panel **e** demonstrating higher intensity of active CCND1-MS2 genes in cells with high nuclear YFP-β-catenin levels (red arrows) compared to cells with low nuclear YFP-β-catenin levels (yellow arrows). Active genes are pseudo-colored using the ImageJ 'Red Hot' look-up table. The fluorescent signal of the active genes was enhanced using ImageJ 'Spot Enhancing Filter 2D'. This enhancement led to the reduced detectability of single mRNAs in this presentation of the image, in order to emphasize the difference in transcriptional activity between low and high levels of nuclear YFP-β-catenin. Bar = 10 μm.

The following figure supplement is available for figure 8:

**Figure supplement 1.** Transcription site intensity levels in living cells following Wnt3a activation.

Notably, our study also provides a temporal view of β-catenin dynamics in single cells under conditions of LiCl activation. Although LiCl is considered a chemical that mimics Wnt activation and increases β-catenin levels in the nucleus, it is obvious that the dynamics, build-up rates and levels of β-catenin in all subcellular compartments were dramatically exaggerated and unregulated in comparison to Wnt activation. This should be taken into account when inferring information regarding Wnt signaling and β-catenin from LiCl treatment.

The Wnt pathway has been implicated in cell cycle regulation, and levels of phosporylated β-catenin oscillate and increase towards mitosis (*Davidson and Niehrs, 2010*; *Hadjihannas et al., 2012*). Examining cells that had undergone mitosis after Wnt activation, did not show a pattern of β-catenin levels in daughter cells, nor did Fucci labeling uncover a cell cycle pattern of β-catenin accumulation following Wnt. This suggests that Wnt-induced nuclear accumulation is not cell cycle dependent.

The propagation of a signal from a membrane receptor to the gene promoter can follow different types of kinetics. Single-cell analysis revealed significant variability in the dynamics of β-catenin nuclear buildup, but also that most cells did finally accumulate the same total level of β-catenin over time. This behavior is quite different than the serum activation pathway that activates β-actin via MAL shuttling (*Kalo et al., 2015*). β-actin transcriptional activation begins less than 5 min after serum addition, and β-actin alleles respond in the same manner and same time-frame; i.e. variability of the response in single cells is low. Hence, some signaling cascades must relay the information rapidly and tightly since this will lead to the translation of a highly required protein, e.g. β-actin, to generate a protein that is required for cell motility in response to environmental sensing (*Kislauskis et al., 1994*; *Latham et al., 1994*). Other pathways such as Wnt/β-catenin may also signal to activate gene expression, but their response emerges much later, probably since the required biological outcomes, such as cell proliferation, require more regulation points. The changes in β-catenin levels in response to Wnt, in several subcellular compartments, indicate that the signaling pathway does not only activate gene expression but is involved in additional processes. Further studies should reveal the exact roles of these subpopulations of β-catenin in response to signal transduction.

## Materials and methods

### Cells and transfections

HEK293 Flp-in CCND1-MS2 cells (*Yunger et al., 2010*) were maintained in Dulbecco's modified Eagle's medium (DMEM, Biological Industries, Israel) containing 10% FBS (HyClone Laboratories, Logan, UT) and hygromycin selection (100 μg/ml; Sigma, Israel). Stable expression of MS2-GFP was obtained by co-transfection of the cells with MS2-GFP (10 μg) and puromycin

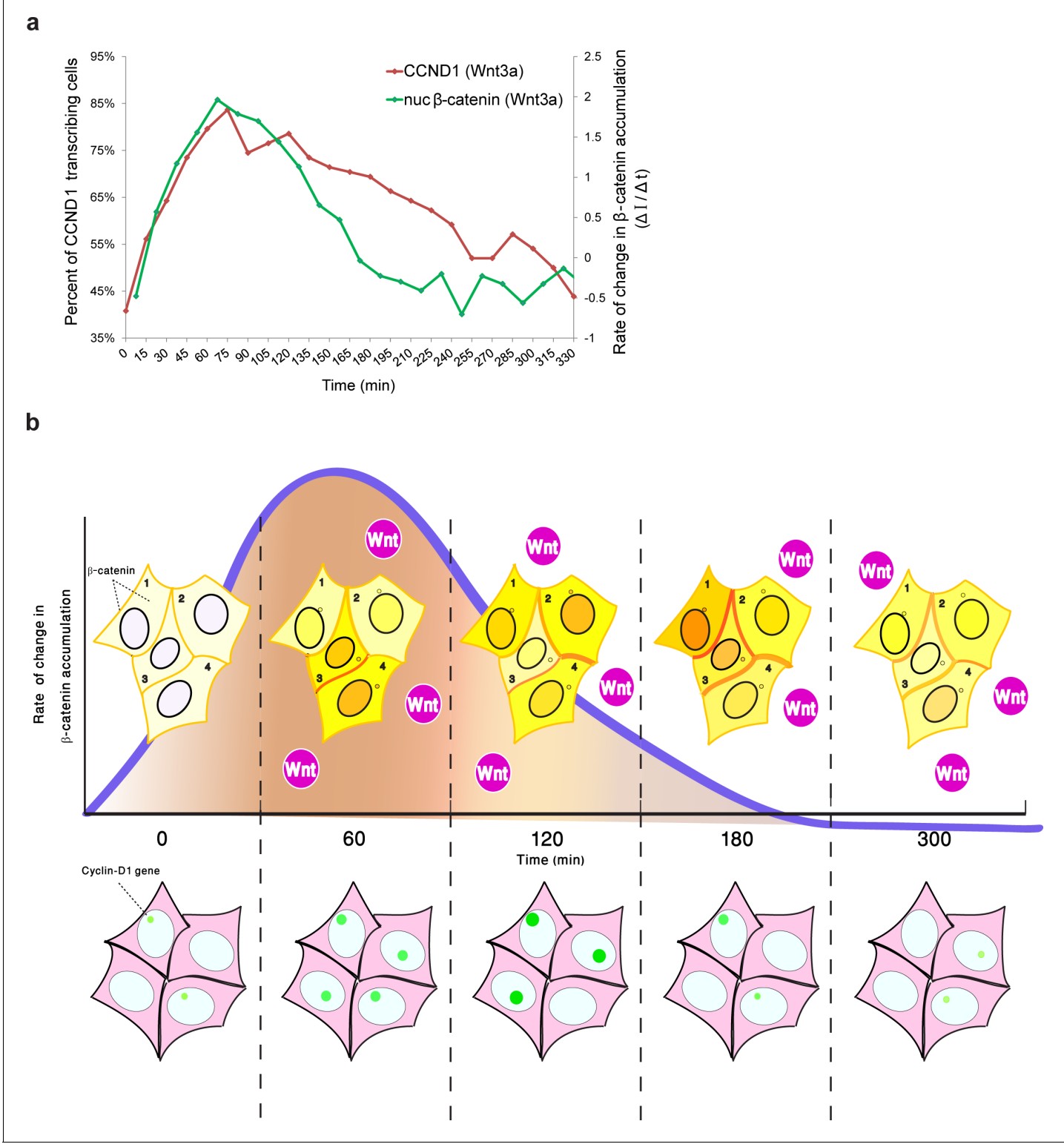

**Figure 9.** Comparing the kinetics of CCND1 transcriptional activation to the dynamics of β-catenin nuclear accumulation rate of change following Wnt signaling in living cells. (a) Plots of the average transcriptional activation kinetics of CCND1-MS2 (red) following Wnt3a activation, compared to the plot of rate of change in β-catenin nuclear accumulation (green). (b) Scheme of the dynamic changes occurring in the studied cell system following Wnt signaling. Top - from left to right: Levels of β-catenin (yellow) in the nucleus are normally low but after the addition of Wnt3a to the medium a significant and rapid increase in the nucleus is observed, peaking after 2–3 hr. β-catenin levels later decline in the nucleus and cytoplasm due to degradation. While this is the average behavior in the population (e.g. cells 1 and 2), when examining individual cells, different dynamics such as
*Figure 9 continued on next page*

*Figure 9 continued*

multiple pulsations (e.g. cell 3) and rapid initial accumulation (e.g. cell 4) are observed. β-catenin levels increase simultaneously at the membrane and at the centrosome. Bottom- β-catenin induces cyclin D1 transcriptional activity (green dot), and modulation of the transcriptional reaction can be observed as the gene reaches higher levels of activity, for longer periods of time. The rate of change in β-catenin accumulation (blue curve, top), rather than the actual levels of β-catenin in the nucleus, correlate with the kinetics of transcriptional activation.

resistance (300 ng) plasmids using calcium phosphate transfection, and selection with puromycin (1 µg/ml; Invivogen, San Diego, CA) and hygromycin (100 µg/ml). Stable expression of YFP-β-catenin (*Krieghoff et al., 2006*) (10 µg) was performed by calcium phosphate transfection, and selection with neomycin (500 µg/ml; Sigma) and hygromycin (100 µg/ml). Cells with very low expression levels were collected by FACS (FACSAria III, BD Biosciences). Transient expression of YFP-β-catenin was performed using PolyJET (SignaGen, Israel).

For generating Wnt3a conditioned medium (CM) and mock CM, L-Wnt-3A and L- mouse fibro-blast cells were grown in DMEM and 10% FBS, and CM was prepared according to the American Tissue Culture Collection (ATCC) instructions (*Shibamoto et al., 1998*). Wnt activation was performed with either Wnt3a-CM or with recombinant human Wnt3a (200 ng/ml; R& D Systems, Minneapolis, MN). Wnt3a-CM or mock-CM were added 1:1 to the volume of the cells medium. Cells were also treated with LiCl (20 mM; Sigma) and MG132 (20 µM; Sigma).

The Fucci system (Clontech, Mountain View, CA) was used for cell cycle phase detection. For G1 phase detection, the pRetroX-G1-Red vector (mCherry-hCdt1) was used, and for S/G2/M phase the pRetroX-SG2M-Cyan vector (AmCyan-hGeminin). The Fucci system, being a viral-based system first required the introduction of the mouse ecotropic retroviral receptor on the membrane surface of HEK293 CCND1-MS2 cells expressing YFP-β-catenin. Transient transfection was performed 24 hr prior to infection using PolyJet transfection with the pBABE ecotropic receptor plasmid (Addgene #10687, Cambridge, MA). This step was performed twice for each infection. After mCherry-hCdt1 infection, mCherry positive cells were collected by FACS and maintained in medium containing puromycin (1 µg/ml; Invivogen). Cells were then transfected with the pBABE ecotropic receptor plasmid and 24 hr post-transfection, the cells infected with AmCyan-hGeminin. Positive cells were collected by FACS and maintained in medium containing neomycin (500 µg/ml) and puromycin (1 µg/ml). For infections, HEK293T cells were maintained in DMEM containing 10% FBS and used to package the Fucci retroviruses, which were collected over a period of three days before infecting the ecotropic HEK293 cells.

## Western blotting

SDS-PAGE and Western blotting were performed as previously described (*Aizer et al., 2008*). Primary antibodies used were mouse anti-β-catenin (BD Transduction Laboratories, cat# 610154, San Jose, CA) and rabbit anti-tubulin (Abcam, Cambridge, MA). The secondary antibody was an HRP-conjugated goat anti-rabbit or anti-mouse IgG (Sigma). Immunoreactive bands were detected by the Enhanced Chemiluminescence kit (ECL, PierceThermo scientific, Waltham, MA). Experiments were performed three times.

## Luciferase assay

HEK293 CCND1-MS2 cells were co-transfected with the cyclin D1 promoter −1745CD1LUC Firefly luciefarse construct (*Albanese et al., 1995*) and either YFP-β-catenin or eYFP-C1 (mock), together with a *Renilla* luciferase construct using PolyJet transfection. 50 ng of each plasmid were used. A luciferase assay was performed after 24 hr using the Dual-Glo Luciferase assay system (Promega, Madison, WI). After standardization with *Renilla* luciferase activity, a relative luciferase activity was obtained and the mean and standard deviation from triplicate wells was calculated. Each experiment was performed three times. YFP-β-catenin (*Krieghoff et al., 2006*) was obtained from Jürgen Behrens (University of Erlangen-Nürnberg).

## Flow cytometry

Cells were harvested and DNA quantification was performed using 5 µg/ml DAPI solution (Sigma). The BD FACSAria III cell sorter was used. For quantifying DNA in fixed cells, we used a 405 nm laser

for excitation and a 450/40 nm bandpass filter for detection. Data were processed and analyzed using FlowJo software. The average quantification of 3 repeated experiments is presented (mean ± sd).

## Immunofluorescence

Cells were grown on coverslips coated by Cell-Tak (BD Biosciences), washed with PBS and fixed for 20 min in 4% PFA. Cells were then permeabilized in 0.5% Triton X-100 for 3 min. After blocking, cells were immunostained for 1 hr with a primary antibody, and after subsequent washes the cells were incubated for 1 hr with secondary fluorescent antibodies. Primary antibodies: mouse anti-β-catenin and rabbit anti-pericentrin (Abcam, cat# ab4448). Secondary antibodies: Alexa488-labeled goat anti-mouse IgG and Alexa594-labeled goat anti-rabbit (Invitrogen, Carlsbad, CA). Nuclei were counter-stained with Hoechst 33342 (Sigma) and coverslips were mounted in mounting medium.

## Fluorescence in situ hybridization

CCND1-MS2 cells were grown on coverslips coated by Cell-Tak (BD Biosciences) and fixed for 20 min in 4% paraformaldehyde, and overnight with 70% ethanol at 4°C. The next day cells were washed with 1x PBS and treated for 2.5 min with 0.5% Triton X-100. Cells were washed with 1x PBS and incubated for 10 min in 40% formamide (4% SSC; Sigma). Cells were hybridized overnight at 37°C in 40% formamide with a specific fluorescently-labeled Cy3 DNA probe (~10 ng probe, 50 mer). The next day, cells were washed twice with 40% formamide for 15 min and then washed for two hours with 1X PBS. Nuclei were counterstained with Hoechst 33342 and coverslips were mounted in mounting medium. The probe for the MS2 binding site was:
CTAGGCAATTAGGTACCTTAGGATCTAATGAACCCGGGAATACTGCAGAC.

## mRNA quantification

3D stacks (0.2 μm steps, 76 or 51 planes) of the total volume of the cells were collected from fixed CCND1-MS2 cells. The 3D stacks were deconvolved and the specific signals of mRNAs were identi-fied (Imaris, Bitplane). mRNA identification was performed in comparison to deconvolved stacks from cells not containing the MS2 integration, which therefore served as background levels of non-specific fluorescence. No mRNAs were identified in control cells. The sum of intensity for each mRNA particle and active alleles was measured in the same cells using Imaris, as previously described (*Yunger et al., 2010*, *2013*). The single mRNA intensities were pooled and the frequent value was calculated. The sum of intensity at the transcription site was divided by the frequent value of a single mRNA. This ratio provided the number of mRNAs associated with the transcription unit from the point of the MS2-region and onwards. As mRNAs should be associated with a polymerase, this number should reflect the maximum number of polymerases engaged with this region. Quantifi-cation and counting experiments were applied to experiments performed on different days.

## Fluorescence microscopy, live-cell imaging and data analysis

Wide-field fluorescence images were obtained using the CellR system based on an Olympus IX81 fully motorized inverted microscope (60X PlanApo objective, 1.42 NA) fitted with an Orca-AG CCD camera (Hamamatsu) driven by the CellR software. Live-cell imaging was carried out using the CellR system with rapid wavelength switching. For time-lapse imaging, cells were plated on glass-bot-tomed tissue culture plates (MatTek, Ashland, MA) coated by Cell-Tak (BD Biosciences) in medium containing 10% FBS at 37°C. The microscope is equipped with an incubator that includes tempera-ture and $CO_2$ control (Life Imaging Services, Reinach, Switzerland). For long-term imaging, several cell positions were chosen and recorded by a motorized stage (Scan IM, Märzhäuser, Wetzlar-Stein-dorf, Germany). In these experiments, HEK293 Flp-in CCND1-MS2 expressing MS2-GFP cells were imaged in 3D (26 planes per time point) every 15 min, at 0.26 μm steps for 6 hr. HEK293 Flp-in CCND1-MS2 cells expressing YFP-β-catenin were imaged in 3D (15 planes per time point) at 0.7 μm steps, every 15 min, up to 18 hr. For presentation of the movies, the 4D image sequences were transformed into a time sequence using the maximum or sum projection options or manually select-ing the in-focus plane using the ImageJ software. Time-lapse data were collected from single cells in several fields and on several days until reaching an appropriate sample size, and then all single-cell data were pooled and either averaged and presented as plots, or presented as single cell data.

## Tracking and data analysis

The intensity of the active transcription sites labeled with MS2-GFP fluorescence in time-lapse movies was corrected for photobleaching using ImageJ, and the 3D movies were transformed to 2D by choosing the in-focus plane in which the intensity of the transcription site is the highest. Movies were manually tracked and the intensity measured for each frame (*Is*). Background from another location in the nucleus (*In*) was subtracted for each frame, and the final intensity was calculated using: $I = Is(t) - In(t)$ and then normalized to the initial intensity.

Measuring the intensity of the YFP-β-catenin signal in the subcellular compartments was performed manually using ImageJ, and background was subtracted from all measurements. When YFP-β-catenin levels were low, DIC images that were acquired in parallel were used for nucleus detection. For measurements of centrosome intensity, the intensity of the centrosome in each frame (*Ic*) was multiplied by the area occupied by the centrosome (*Ac*): $I = Ic(t) *Ac(t)$. For membrane intensity, a sum projection of the 3D movies was used.

Intensity was normalized either to the initial frame or to the highest intensity measured. The values of the nucleus/cytoplasm (N/C) ratio of YFP-β-catenin were obtained by division of the YFP-β-catenin intensity levels measured. Correlation coefficient values were calculated by comparing the intensity of β-catenin over time between all possible pairs of sub-cellular compartments, from Wnt activation onset. Values of rate of change ($\Delta I/\Delta t$) in YFP-β-catenin in the sub-cellular compartments over time were obtained by measuring the intensity difference ($\Delta I$) between two consecutive time points divided by the time difference ($\Delta t$) between the two time points:

$$\frac{\Delta I}{\Delta t}_{(t)n+\frac{1}{2}} = \frac{I_{(t)n+1} - I_{(t)n}}{t_{n+1} - t_n}$$

## FRAP and FLIP

FRAP and FLIP experiments were performed using a 3D-FRAP system (Photometrics) built on an Olympus IX81 microscope (636 Plan-Apo, 1.4 NA) equipped with an EM-CCD (Quant-EM, Roper), 491 nm laser, Lambda DG-4 light source (Sutter), XY and Z stages (Prior), and driven by MetaMorph (Molecular Devices). Experiments were performed at 37°C with 5% $CO_2$ using a live-cell chamber system (Tokai). For each acquisition, YFP-β-catenin was bleached using the 491 nm laser. Six pre-bleach images were acquired. In FRAP, post-bleach images were acquired every 0.8 s for 80 s in the cytoplasm and the nucleus, every 1 s for 2 min in adherens junctions, every 0.4 s for 40 s at the centrosome, and every 1.5 s for 8 min to measure nuclear import and export rates. In FLIP, images were acquired every 1.9 s for 280 s in the cytoplasm and the nucleus. The experiments were analyzed using ImageJ macros previously described (*Aizer et al., 2008*). Data from at least 10 experiments for each cell line were collected and the averaged FRAP and FLIP measurements were fitted by Matlab with a double exponential model:

$$I(t) = \alpha_1 * exp(-\tau_1 * t) + \alpha_2 * exp(-\tau_2 * t) + c$$

Where *t* = 0 is the time immediately after photobleaching.

$t_{0.5}$ was defined as time where $I(t = t_{0.5}) = \frac{I(t=\infty)}{2}$.

## Modeling β-catenin dynamics

We used a simple model for describing β-catenin concentration (*C*) dynamics in the nucleus based on the data presented in the plot from *Figure 2c*:

$$\frac{dC}{dt} = P(t) - \alpha(t)C$$

Where $\alpha$ is the time dependent degradation rate, and *P* is the time dependent production rate. Both rates are allowed to change when $t = T$:

$$\alpha(t) = \begin{cases} \alpha_1; & for \quad t \leq T \\ \alpha_2; & for \quad t > T \end{cases}$$

$$P(t) = \begin{cases} P_1; & for \quad t \leq T \\ P_2; & for \quad t > T \end{cases}$$

The solution is:

$$C(t) = \begin{cases} \left[ C(0) - \frac{P_1}{\alpha_1} \right] * e^{-\alpha_1 * t} + \frac{P_1}{\alpha_1}; & for \; t \leq T \\ \left[ C(T) - \frac{P_2}{\alpha_2} \right] * e^{-\alpha_2 * (t-T)} + \frac{P_2}{\alpha_2}; & for \; t > T \end{cases}$$

Where:

$$C(T) = \left[ C(0) - \frac{P_2}{\alpha_1} \right] * e^{-\alpha_1 * t} + \frac{P_2}{\alpha_1}$$

We fit the model by minimizing the sum of the squares of the residuals with the function 'fmincon' in MATLAB using the 'active-set' algorithm.

## Statistical analysis

Two tailed t-test was performed in the following experiments: Quantitative FISH, Luciferase assay, the N/C ratio of YFP-β-catenin and live cell analysis. A Mann–Whitney test was performed in FRAP and FLIP experiments (*Supplementary file 1*).

## Acknowledgements

We thank Yinon Ben-Neriah (Hebrew University Medical School) for the L-Wnt-3A and L- mouse fibroblast cells and Jürgen Behrens (University of Erlangen-Nürnberg) for YFP-β-catenin. We thank Bezalel Shav-Tal for proofreading the manuscript. This work was supported by the European Research Council and the Israel Cancer Research Fund (YST).

## Additional information

### Funding

| Funder | Author |
| --- | --- |
| European Research Council | Yaron Shav-Tal |
| Israel Cancer Research Fund | Yaron Shav-Tal |

The funders had no role in study design, data collection and interpretation, or the decision to submit the work for publication.

### Author contributions

PK, Conception and design, Acquisition of data, Analysis and interpretation of data, Drafting or revising the article; SEH, Acquisition of data, Analysis and interpretation of data; IK, Analysis and interpretation of data, Drafting or revising the article; JS, NK, SY, Acquisition of data; YS-T, Conception and design, Analysis and interpretation of data, Drafting or revising the article

### Author ORCIDs

Yaron Shav-Tal, http://orcid.org/0000-0002-8017-948X

## Additional files

### Supplementary files

• Supplementary file 1. Statistical analysis performed in this study. (a) The statistical significance p values (t test) at each time point for the percentage of cells showing an active CCND1-MS2 gene (refers to *Figure 1b*) between control and Wnt3a-treated cells. (b-f) Mann-Whitney test for comparison between two independent FRAP/FLIP experiments. A statistical comparison between all datasets of two individual FRAP/FLIP experiments are depicted in each plot and are illustrated as a

single red circle which marks the p-value (y axis) for all intensity values measured for each time point (x axis). The top and bottom dotted lines indicate where p-value equals 0.05. (b) Statistically significant difference between the FRAP dynamics of YFP-β-catenin in the nucleus under Wnt3a treatment versus overexpression of YFP-β-catenin that enters the nucleus without signal, and (c) between the FRAP and (d) FLIP import and export dynamics (refers to *Figure 2—figure supplement 1*). (e) No statistically significant difference between YFP-β-catenin at the cell membrane between mock-treated and Wnt3a-treated cells (refers to *Figure 5—figure supplement 1*). (f) No statistically significant difference between YFP-β-catenin at the cell membrane between mock-treated and LiCl-treated cells (refers to *Figure 5—figure supplement 1*).

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
