## [Decision Letter]

Thank you for submitting your article "Signaling to gene induction dynamics yield varying reactions in single cells forming a coordinated population response" for consideration by *eLife*. Your article has been reviewed by two peer reviewers, and the evaluation has been overseen by a Reviewing Editor and Kevin Struhl as the Senior Editor. The reviewers have opted to remain anonymous.

The reviewers have discussed the reviews with one another and the Reviewing Editor has drafted this decision to help you prepare a revised submission. There agree a considerable number of requirements for revisions. If you are not able to submit the revised version within about two months, we will consider the manuscript as a resubmission rather than a revision.

Summary:

The authors investigate the intracellular dynamics of β-catenin, and the transcriptional response of a downstream promoter, following Wnt signaling. Both aspects are studied in individual live cells, using a YFP‐tagged version of the protein and an MS2-GFP labeled version of the RNA. Mapping the "signaling-to-gene-induction dynamics" (as promised in the title) is indeed an important goal.

Essential revisions:

1) Combine the two reporter systems in the same cell. This is essential because there is no single-cell measurement of how β-catenin dynamics and transcription are related, which is the whole premise of the paper.

2) Quantify the kinetics of nuclear import/export e.g. by measuring the rate of change in nuclear-to-cytoplasmic ratio (or difference).

3) Simplify the title to better reflect the main findings of the manuscript.

4) The manuscript should be simplified, especially the Introduction.

5) Demonstrate at the single cell level whether in a cell nucleus accumulating YFP-β-catenin, an increased CCND1 transcription can be observed at the same time and to same extent.

6) Use a cell line in which the FP tag is knocked in to the wild type β-catenin locus. At minimum, the two-fold total amount of β-catenin in the cells has to be clearly stated.

7) Figure 2—figure supplement 1: Plot normalized fluorescence intensity in the cytoplasm for the same period as for the recovery time in the nucleus. If there is no import, it should be a flat line.

*Reviewer #1:*

The authors investigate the intracellular dynamics of β-catenin, and the transcriptional response of a downstream promoter, following Wnt signaling. Both aspects are studied in individual live cells, using a YFP‐tagged version of the protein and an MS2-GFP labeled version of the RNA. Mapping the "signaling-to-gene-induction dynamics" (as promised in the title) is indeed an important goal. I was therefore excited about reading the paper. Unfortunately, I ended up rather disappointed, since I felt that the promise was not fulfilled:

1) Most conspicuously, the authors never actually combine the two reporter systems in the same cell. Thus, there is no single-cell measurement of how β-catenin dynamics and transcription are related, which a-priori seemed like the whole premise of the paper. Even in the Discussion, the authors still state that their approach "combines the real‐time measurements of the signaling protein factor together with gene activity readout in individual living cells". How so?

2) There are other instances where the data doesn't match up to the verbal statements. For example, the authors repeatedly mention "the kinetics of nuclear clearance", "shuttling in and out of the nucleus", or the "very slow egress" of the protein, but the kinetics of nuclear import/export are never quantified, e.g. by measuring the rate of change in nuclear-to-cytoplasmic ratio (or difference). Why is that?

3) The manuscript is also overly long. The Introduction, for example, reads like part of a review article-it broadly surveys the field rather than leading the narrative to the specific questions that will be addressed in the work. Elsewhere, there are multiple occurrences of ambiguous statements. Even the title, to me, is ambiguous unless some hyphens and commas are added. Thus, significant re-writing should be done if the manuscript is accepted.

*Reviewer #2:*

1) The title needs to be simplified and it should better reflect the main findings of the manuscript.

2) The manuscript would benefit from simplification. It contains too much data that will prevent the general reader from following the flow of the results and their importance. For example is Figure 8 absolutely necessary?

3) The authors aim is to understand by which mechanism a signalling pathway acting at the cell population level is creating variability in responses at the single cell level. However, this question is not fully answered in this manuscript (see also comment 7).

4) Figure 1: The authors carry out experiments to verify whether the YFP-β-catenin behaves and is localised similarly to the endogenous protein. Their assays seem to indicate that the addition of Wnt3a has the same effect on the expression of YFP-β-catenin as it has on the endogenous β-catenin. In contrast, the addition of LiCl seems to induce a higher expression of YFP-β-catenin than that of the endogenous β-catenin. (The error bars in the quantification histogram of the LiCl WB are quite big, are these significant?). Following these experiments the authors conclude that the exogenously expressed YFP-β-catenin can be used in the further experiments as it behaves similarly to the endogenous protein. Nevertheless, in these YFP-β-catenin cells the total amount of β-catenin is the double than that in wild type cells and this β-catenin overexpression could lead to misinterpretations.

5) Along the same lines the authors write, "YFP-catenin is equal to endogenous levels". As mentioned above, this means that there is a 2x fold increase in the total amount of β-catenin protein that can influence the results obtained with YFP-β-catenin and especially by those studying the target gene activation. At present the authors give the impression that the study is performed in a system in which the total β-catenin is at endogenous levels. It would be better to use a cell line in which the FP tag is knocked in to the wild type β-catenin locus. At minimum, the two-fold total amount of β-catenin in the cells has to be clearly stated.

6) Figure 2—figure supplement 1: Normalized fluorescence intensity in the cytoplasm should be plotted for the same period as for the recovery time in the nucleus. If there is no import, it should be a flat line.

7) Figure 9: In this last figure the authors present a model that is supposed to summarize their findings. While in a cell population they can show that CCND1 transcription is lasting longer than YFP-β-catenin accumulation in the nuclei of the cells, they fail to demonstrate at the single cell level whether in a YFP-β-catenin accumulating cell nucleus an increased CCND1 transcription can be observed at the same time and to same extent. This is a potential weakness of the manuscript that is also reflected in the model, where one does not know in which β-catenin-containing/expressing nucleus the CCND1 transcription is increased, when comparing the cells drawn in the upper row and in the lower row. Would it be possible to quantify the CCND1 transcription in the YFP-β-catenin containing nuclei and correlate the CCND1 expression with the YFP-β-catenin signal in a single cell based measurement series?

[Editors' note: further revisions were requested prior to acceptance, as described below.]

Thank you for submitting your article "Quantifying β-catenin subcellular dynamics and gene expression kinetics during Wnt signaling in single living cells" for consideration by *eLife*. Your article has been reviewed by two peer reviewers, and the evaluation has been overseen by a Reviewing Editor and Kevin Struhl as the Senior Editor. The reviewers have opted to remain anonymous.

The reviewers have discussed the reviews with one another and the Reviewing Editor has drafted this decision to help you prepare a revised submission.

While both reviewers agree that the manuscript is improved, in the ensuing discussion they both also agree that the two reporters in the same cell is a requirement not met. One raises the possibility to modify your conclusions to accommodate this omitted data, the other feels that it is necessary but would agree to oversee a modified manuscript that does not depend on this experiment.

We all feel the manuscript would be much stronger with the experiment than without, but it's up to you how you want to handle this revision.

*Reviewer #1:*

I do not think that the authors have improved the paper sufficiently to warrant publication. Specifically:

1) The main request, to "combine the two reporter systems in the same cell", was not fully met. What the authors did was simultaneously measure β-catenin (protein) and CCND1 (RNA) in FIXED cells. While this measurement can potentially provide useful information, it falls short of the goal of detecting, and correlating, both species in real-time in live cells.

The way the authors analyze the protein/RNA data also seems strange. They count the total number of CCND1-MS2 RNA molecules per cell and correlate it with the nuclear level of β-catenin (Figure 8). However, the total cellular count represents the accumulation of RNA over the lifetime of RNA molecules, which could be hours. To probe the regulatory relation between protein concentration and transcriptional activity, the authors should have instead examined the probability of active transcription, or the amount of nascent RNA, just as they did earlier in Figure 1.

2) The text needs to be edited and proofread. Proofreading by an English speaker would have helped in some instances (single/plural, misuse of verbs, etc.), but in other instances this is a matter of ambiguous and confusing statements. The editorial team will have to address this problem if the manuscript is chosen for publication. To give just one example, the title, "Quantifying β-catenin subcellular dynamics and gene expression kinetics" implies that both the subcellular dynamics and the gene expression kinetics are those of β-catenin, but this is not actually the case.

*Reviewer #2:*

The authors answered (more or less) my comments and made most of the suggested improvements. The manuscript is considerably improved.

---

## [Author Response]

*Essential revisions:*

*1) Combine the two reporter systems in the same cell. This is essential because there is no single-cell measurement of how β-catenin dynamics and transcription are related, which is the whole premise of the paper.*

This point was raised by both reviewers (also no. 6 below), namely, to show that β-catenin levels lead to an increase in the CCND1 transcript population in the same cell. In order to demonstrate in single cells within a population that β-catenin nuclear accumulation correlates with high CCND1 mRNA levels, we counted single mRNA molecules in single cells expressing the YFP-β-catenin protein using RNA FISH, after Wnt addition for 2 hours. We show that cells in the population that have high YFP-β-catenin nuclear accumulation also have significantly higher numbers of the mRNA. We also perform correlation analysis showing the connection between YFP-β-catenin and mRNA numbers. This now appears in a new Figure 8 (after reviewer 2 suggested we remove the old Figure 8).

*2) Quantify the kinetics of nuclear import/export e.g. by measuring the rate of change in nuclear-to-cytoplasmic ratio (or difference).*

This suggestion was useful since we now can show the difference between YFP-β-catenin import and export in living cells. The import FRAP experiment (Figure 2—figure supplement 1) now also shows the reduction in the cytoplasm together with the increase in the nucleus (as requested by reviewer 2). The figure has an addition: a) of a complementary FRAP experiment in the cytoplasm; and b) of a FLIP experiment either in the nucleus or in the cytoplasm. Both experiments show that YFP-β-catenin import is more rapid than export.

3) Simplify the title to better reflect the main findings of the manuscript.

Following remarks by both reviewers we changed the title to:

“Quantifying β-catenin subcellular dynamics and gene expression kinetics during Wnt signaling in single living cells”. We acknowledge that the previous title was grammatically problematic as mentioned by reviewer 1, but that was due to the 120-character cap. We hope the new title gives a better impression of the study as a whole.

*4) The manuscript should be simplified, especially the Introduction.*

We have made large efforts to cut down and tighten the text. The Introduction has been simplified and extensively shortened, leaving only the relevant background for the study. We have also focused the Discussion section and removed less important issues. We removed one supplementary figure (56) and also removed most of Figure 8 as suggested by reviewer 2. (New Figure 8 contains new the examination of β-catenin and CCND1 RNA FISH in the same cell). In the Results sections we tried to cut down where possible, and some sections describing for example fold changes in figures were simplified, so to only describe the data in general, and the readers can decide if they want to delve into the details. Some text from the Results section was moved to the Discussion, so not to be repeated twice.

*5) Demonstrate at the single cell level whether in a cell nucleus accumulating YFP-β-catenin, an increased CCND1 transcription can be observed at the same time and to same extent.*

This point was raised by both reviewers (also no. 1 above), namely, to show that β-catenin levels lead to an increase in the CCND1 transcript population in the same cell. In order to demonstrate in single cells within a population that β-catenin nuclear accumulation correlates with high CCND1 mRNA levels, we counted single mRNA molecules in single cells expressing the YFP-β-catenin protein using RNA FISH, after Wnt addition for 2 hours. We show that cells in the population that have high YFP-β-catenin nuclear accumulation also have significantly higher numbers of the mRNA. We also perform correlation analysis showing the connection between YFP-β-catenin and mRNA numbers. This now appears in a new Figure 8 (after reviewer 2 suggested we remove the old Figure 8).

*6) Use a cell line in which the FP tag is knocked in to the wild type β-catenin locus. At minimum, the two-fold total amount of β-catenin in the cells has to be clearly stated.*

We do not have a cell line with the FP knocked in. We started this project some years before CRISPR appeared on the scene. We now mention specifically that YFP-β-catenin is overexpressed above the background of the endogenous protein. We also quantified this according to the Western blots in order to give an actual number, and find that YFP-β-catenin protein levels are 0.8x fold (almost double) over the endogenous levels of β-catenin protein. This is mentioned twice, in the Results and in the Discussion. Figure 1—figure supplement 1 now also includes flow cytometry analysis showing that the overexpression of YFP-β-catenin does not change the cell cycle profile of the cells in the population.

*7) Figure 2—figure supplement 1: Plot normalized fluorescence intensity in the cytoplasm for the same period as for the recovery time in the nucleus. If there is no import, it should be a flat line.*

That was a good idea since the data are contained in the movie. We therefore analyzed the data for the cytoplasm as suggested and indeed the cytoplasmic fluorescence decreases in correlation to the increase in nuclear accumulation.

*Reviewer #1:*

*The authors investigate the intracellular dynamics of β-catenin, and the transcriptional response of a downstream promoter, following Wnt signaling. Both aspects are studied in individual live cells, using a YFP‐tagged version of the protein and an MS2-GFP labeled version of the RNA. Mapping the "signaling-to-gene-induction dynamics" (as promised in the title) is indeed an important goal. I was therefore excited about reading the paper. Unfortunately, I ended up rather disappointed, since I felt that the promise was not fulfilled:*

*1) Most conspicuously, the authors never actually combine the two reporter systems in the same cell. Thus, there is no single-cell measurement of how β-catenin dynamics and transcription are related, which a-priori seemed like the whole premise of the paper. Even in the Discussion, the authors still state that their approach "combines the real‐time measurements of the signaling protein factor together with gene activity readout in individual living cells". How so?*

This point was raised by both reviewers, namely, to show that β-catenin levels lead to an increase in the CCND1 transcript population in the same cell. In order to demonstrate in single cells within a population that β-catenin nuclear accumulation correlates with high CCND1 mRNA levels, we counted single mRNA molecules in single cells expressing the YFP-β-catenin protein using RNA FISH, after Wnt addition for 2 hours. We show that cells in the population that have high YFP-β-catenin nuclear accumulation also have significantly higher numbers of the mRNA. We also perform correlation analysis showing the connection between YFP-β-catenin and mRNA numbers. This now appears in a new Figure 8 (after reviewer 2 suggested we remove the old Figure 8). In the shortening of the Discussion this sentence was part of a section removed and we have made efforts to more clearly phrase what we are showing.

*2) There are other instances where the data doesn't match up to the verbal statements. For example, the authors repeatedly mention "the kinetics of nuclear clearance", "shuttling in and out of the nucleus", or the "very slow egress" of the protein, but the kinetics of nuclear import/export are never quantified, e.g. by measuring the rate of change in nuclear-to-cytoplasmic ratio (or difference). Why is that?*

First, we have tried to be more careful in our statements so that they better describe the actual data. With regard to issues of dynamics of import and export of β-catenin we have made the following changes and added new data. The import FRAP experiment (Figure 2—figure supplement 1) now also shows the reduction in the cytoplasm together with the increase in the nucleus (as requested by reviewer 2). The figure has an addition: a) of a complementary FRAP experiment in the cytoplasm; and b) of a FLIP experiment either in the nucleus or in the cytoplasm. Both experiments show that YFP-β-catenin import is more rapid than export. Furthermore, we have fit the curve in Figure 2 that describes the accumulation and reduction dynamics of YFP-β-catenin in the cell population, and modeled the data. Using this we can now better describe the accumulation phase that does not include degradation, as well as the clearance phase in which degradation kicks in.

*3) The manuscript is also overly long. The Introduction, for example, reads like part of a review article-it broadly surveys the field rather than leading the narrative to the specific questions that will be addressed in the work. Elsewhere, there are multiple occurrences of ambiguous statements. Even the title, to me, is ambiguous unless some hyphens and commas are added. Thus, significant re-writing should be done if the manuscript is accepted.*

We have made large efforts to cut down and tighten the text. The Introduction has been simplified and extensively shortened, leaving only the relevant background for the study. We have also focused the Discussion section and removed less important issues. We removed one supplementary figure (56) and also removed most of Figure 8 as suggested by reviewer 2. (New Figure 8 contains new the examination of β-catenin and CCND1 RNA FISH in the same cell). In the Results sections we tried to cut down where possible, and some sections describing for example fold changes in figures were simplified, so to only describe the data in general, and the readers can decide if they want to delve into the details. Some text from the Results section was moved to the Discussion, so not to be repeated twice.

Following remarks by both reviewers we changed the title to: “Quantifying β-catenin subcellular dynamics and gene expression kinetics during Wnt signaling in single living cells”. We acknowledge that the previous title was grammatically problematic as mentioned by reviewer 1, but that was due to the 120-character cap. We hope the new title gives a better impression of the study as a whole.

*Reviewer #2:*

*1) The title needs to be simplified and it should better reflect the main findings of the manuscript.*

See previous response.

*2) The manuscript would benefit from simplification. It contains too much data that will prevent the general reader from following the flow of the results and their importance. For example is Figure 8 absolutely necessary?*

We have made large efforts to cut down and tighten the text. The Introduction has been simplified and extensively shortened, leaving only the relevant background for the study. We have also focused the Discussion section and removed less important issues. We removed one supplementary figure (56) and also removed most of Figure 8 as suggested by reviewer 2. (New Figure 8 contains new the examination of β-catenin and CCND1 RNA FISH in the same cell). In the Results sections we tried to cut down where possible, and some sections describing for example fold changes in figures were simplified, so to only describe the data in general, and the readers can decide if they want to delve into the details. Some text from the Results section was moved to the Discussion, so not to be repeated twice.

*3) The authors aim is to understand by which mechanism a signalling pathway acting at the cell population level is creating variability in responses at the single cell level. However, this question is not fully answered in this manuscript (see also comment 7).*

We address this point below, in comment 7.

*4) Figure 1: The authors carry out experiments to verify whether the YFP-β-catenin behaves and is localised similarly to the endogenous protein. Their assays seem to indicate that the addition of Wnt3a has the same effect on the expression of YFP-β-catenin as it has on the endogenous β-catenin. In contrast, the addition of LiCl seems to induce a higher expression of YFP-β-catenin than that of the endogenous β-catenin. (The error bars in the quantification histogram of the LiCl WB are quite big, are these significant?).*

We checked this, and there is no significant difference between endogenous and YFP-β-catenin in the LiCl columns nor between the Wnt columns, in Figure 1. We now mention this in the legend.

*Following these experiments the authors conclude that the exogenously expressed YFP-β-catenin can be used in the further experiments as it behaves similarly to the endogenous protein. Nevertheless, in these YFP-β-catenin cells the total amount of β-catenin is the double than that in wild type cells and this β-catenin overexpression could lead to misinterpretations.*

We answer this below, as the issue of the amount of exogenous β-catenin is also raised in the next comment.

*5) Along the same lines the authors write, "YFP-catenin is equal to endogenous levels". As mentioned above, this means that there is a 2x fold increase in the total amount of β-catenin protein that can influence the results obtained with YFP-β-catenin and especially by those studying the target gene activation. At present the authors give the impression that the study is performed in a system in which the total β-catenin is at endogenous levels. It would be better to use a cell line in which the FP tag is knocked in to the wild type β-catenin locus. At minimum, the two-fold total amount of β-catenin in the cells has to be clearly stated.*

We do not have a cell line with the FP knocked in. We started this project some years before CRISPR appeared on the scene. We now mention specifically that YFP-β-catenin is overexpressed above the background of the endogenous protein. We also quantified this according to the Western blots in order to give an actual number, and find that YFP-β-catenin protein levels are 0.8x fold (almost double) over the endogenous levels of β-catenin protein. This is mentioned twice, in the Results and in the Discussion. Figure 1—figure supplement 1 now also includes flow cytometry analysis showing that the overexpression of YFP-β-catenin does not change the cell cycle profile of the cells in the population.

*6) Figure 2—figure supplement 1: Normalized fluorescence intensity in the cytoplasm should be plotted for the same period as for the recovery time in the nucleus. If there is no import, it should be a flat line.*

That was a good idea since the data are contained in the movie. We therefore analyzed the data for the cytoplasm as suggested and indeed the cytoplasmic fluorescence decreases in correlation to the increase in nuclear accumulation.

*7) Figure 9: In this last figure the authors present a model that is supposed to summarize their findings. While in a cell population they can show that CCND1 transcription is lasting longer than YFP-β-catenin accumulation in the nuclei of the cells, they fail to demonstrate at the single cell level whether in a YFP-β-catenin accumulating cell nucleus an increased CCND1 transcription can be observed at the same time and to same extent. This is a potential weakness of the manuscript that is also reflected in the model, where one does not know in which β-catenin-containing/expressing nucleus the CCND1 transcription is increased, when comparing the cells drawn in the upper row and in the lower row. Would it be possible to quantify the CCND1 transcription in the YFP-β-catenin containing nuclei and correlate the CCND1 expression with the YFP-β-catenin signal in a single cell based measurement series?*

This point was raised by both reviewers, namely, to show that β-catenin levels lead to an increase in the CCND1 transcript population in the same cell. In order to demonstrate in single cells within a population that β-catenin nuclear accumulation correlates with high CCND1 mRNA levels, we counted single mRNA molecules in single cells expressing the YFP-β-catenin protein using RNA FISH, after Wnt addition for 2 hours. We show that cells in the population that have high YFP-β-catenin nuclear accumulation also have significantly higher numbers of the mRNA. We also perform correlation analysis showing the connection between YFP-β-catenin and mRNA numbers. This now appears in a new Figure 8 (after reviewer 2 suggested we remove the old Figure 8). Furthermore regarding what is driving the differences between the response in different cells, we have fit the curve in Figure 2 that describes the accumulation and reduction dynamics of YFP-β-catenin in the cell population, and modeled the data. Using this we can now better describe the accumulation phase that does not include degradation, as well as the clearance phase in which degradation kicks in.

We therefore suggest that the balance between accumulation and degradation is going to affect the outcome in each cell. The Kirschner group has shown (Science 2012) that Wnt does not completely abolish the activity of destruction complex. So we suggest that if the total levels of accumulation are similar in most cells (integral analysis) then is should be the level of inhibition of the destruction complex that varies in each cell and that determines the response.

[Editors' note: further revisions were requested prior to acceptance, as described below.]

*Reviewer #1:*

*I do not think that the authors have improved the paper sufficiently to warrant publication. Specifically:*

*1) The main request, to "combine the two reporter systems in the same cell", was not fully met. What the authors did was simultaneously measure β-catenin (protein) and CCND1 (RNA) in FIXED cells. While this measurement can potentially provide useful information, it falls short of the goal of detecting, and correlating, both species in real-time in live cells.*

*The way the authors analyze the protein/RNA data also seems strange. They count the total number of CCND1-MS2 RNA molecules per cell and correlate it with the nuclear level of β-catenin (Figure 8). However, the total cellular count represents the accumulation of RNA over the lifetime of RNA molecules, which could be hours. To probe the regulatory relation between protein concentration and transcriptional activity, the authors should have instead examined the probability of active transcription, or the amount of nascent RNA, just as they did earlier in Figure 1.*

The issue raised in both revisions was the request to demonstrate β-catenin dynamics and cyclin D1 transcription in the same cell, preferably in a living cell. Obviously, we have tried repeatedly over the years to get this to work in living cells in two colors, but to no avail. We attempted different approaches, and decided to remain with the cell clone expressing the YFP-β-catenin protein so not to begin characterizing another cell clone, but this meant that we could not co-express the GFP-MS2 protein in this clone together with YFP-β-catenin due to the GFP/YFP overlap. Following the request during the first revision, we increased our efforts and began purchasing many types of transfection reagents, and started cloning new versions of either CFP or mCherry MS2-CP, in order to obtain a stable clone with two colors. However, we did not obtain any stable clones by the end of the first revision. Transient transfections did not produce any useful data either because of inefficiency of transfection under these conditions and typical overexpression of the MS2-CP, which masked the transcription site of the single gene. Therefore, to obtain reliable and comparable data we required a stable clone. We have continued trying to generate stable clones with additional types of transfection reagents we purchased during then second revision cycle. We also obtained other MS2-CP plasmids, but all these efforts have been unsuccessful. Perhaps the cells are not very receptacle to the stable integration of additional constructs, because they have already been grown with 3 types of antibiotics. The bottom line is that despite many efforts and financial investment, we have not been able to establish this analysis in two colors in living cells.

But we have performed the analysis of β-catenin and CCND1 mRNA together in fixed cells (Figure 8), and feel that this provides the basic data expected in accordance with the separate live-cell experiments, even though not measured simultaneously in living cells as we would have liked to show. Following the criticism by reviewer 1 in the last revision cycle, we have added to the quantification also the number of nascent mRNAs in each cell. The original logic for presenting the cellular mRNAs was that we thought this quantification presents the accumulative response to Wnt over several hours that can be extracted from a static picture, whereas looking at the levels of gene activity (nascent mRNAs) in some cases might not always give the right impression because the gene might have turned off exactly when the image was acquired. Still, we agree that quantifying the nascent mRNAs in parallel to β-catenin levels gives a complete picture of all parameters that are responding to the Wnt signal, and now present all the data in Figure 8.

Finally, we have made sure not to imply in the manuscript that the two color analysis was performed in living cells, and have: a) not mentioned β-catenin in the part that describes cyclin D1 transcription in living cells, as not give the wrong impression; b) mentioned explicitly that the RNA quantification in correlation to β-catenin accumulation was done in fixed cells; and c) added a sentence in the Discussion that says it outright: “Although we were unable to examine YFP-β-catenin dynamics and CCND1-MS2 transcription activity simultaneously in living cells, by integrating the measurements of CCND1 transcriptional activity with the measured dynamics of β-catenin nuclear accumulation from the separate experiments, we found that…”.

2) The text needs to be edited and proofread. Proofreading by an English speaker would have helped in some instances (single/plural, misuse of verbs, etc.), but in other instances this is a matter of ambiguous and confusing statements. The editorial team will have to address this problem if the manuscript is chosen for publication. To give just one example, the title, "Quantifying β-catenin subcellular dynamics and gene expression kinetics" implies that both the subcellular dynamics and the gene expression kinetics are those of β-catenin, but this is not actually the case.

Grammar – the text has been reviewed and corrected by an academic.

Our revised manuscript is now titled “Quantifying β-catenin subcellular dynamics and cyclin D1 mRNA transcription during Wnt signaling in single living cells”, following reviewer 1’s comments.